# Calcium and bicarbonate signaling pathways have pivotal, resonating roles in matching ATP production to demand

**Maura Greiser[1,2,3†], Mariusz Karbowski[1,4,5†], Aaron David Kaplan[1,6], Andrew Kyle Coleman[1,2], Nicolas Verhoeven[1,4], Carmen A Mannella[1,2], W Jonathan Lederer[1,2,5], Liron Boyman[1,2,5]\***

[1]Center for Biomedical Engineering and Technology, University of Maryland School of Medicine, Baltimore, United States; [2]Department of Physiology, University of Marylan School of Medicine, Baltimore, United States; [3]Claude D. Pepper Older Americans Independence Center, University of Maryland School of Medicine, Baltimore, United States; [4]Department of Biochemistry and Molecular Biology, University of Maryland School of Medicine, Baltimore, United States; [5]Marlene and Stewart Greenebaum Comprehensive Cancer Center, University of Maryland Baltimore School of Medicine, Baltimore, United States; [6]Division of Cardiovascular Medicine, Department of Medicine, University of Maryland School of Medicine, Baltimore, United States

**\*For correspondence:**
Lboyman@som.umaryland.edu

[†]These authors contributed equally to this work

**Competing interest:** The authors declare that no competing interests exist.

**Abstract** Mitochondrial ATP production in ventricular cardiomyocytes must be continually adjusted to rapidly replenish the ATP consumed by the working heart. Two systems are known to be critical in this regulation: mitochondrial matrix $Ca^{2+}$ ($[Ca^{2+}]_m$) and blood flow that is tuned by local cardiomyocyte metabolic signaling. However, these two regulatory systems do not fully account for the physiological range of ATP consumption observed. We report here on the identity, location, and signaling cascade of a third regulatory system -- $CO_2$/bicarbonate. $CO_2$ is generated in the mitochondrial matrix as a metabolic waste product of the oxidation of nutrients. It is a lipid soluble gas that rapidly permeates the inner mitochondrial membrane and produces bicarbonate in a reaction accelerated by carbonic anhydrase. The bicarbonate level is tracked physiologically by a bicarbonate-activated soluble adenylyl cyclase (sAC). Using structural Airyscan super-resolution imaging and functional measurements we find that sAC is primarily inside the mitochondria of ventricular cardiomyocytes where it generates cAMP when activated by bicarbonate. Our data strongly suggest that ATP production in these mitochondria is regulated by this cAMP signaling cascade operating within the inter-membrane space by activating local EPAC1 (**E**xchange **P**rotein directly **A**ctivated by **c**AMP) which turns on Rap1 (Ras-related protein-1). Thus, mitochondrial ATP production is increased by bicarbonate-triggered sAC-signaling through Rap1. Additional evidence is presented indicating that the cAMP signaling itself does not occur directly in the matrix. We also show that this third signaling process involving bicarbonate and sAC activates the mitochondrial ATP production machinery by working independently of, yet in conjunction with, $[Ca^{2+}]_m$-dependent ATP production to meet the energy needs of cellular activity in both health and disease. We propose that the bicarbonate and calcium signaling arms function in a resonant or complementary manner to match mitochondrial ATP production to the full range of energy consumption in ventricular cardiomyocytes.

## Editor's evaluation

Cardiac function is critically dependent on the homeostatic control of ATP so enough energy is provided for a given task and previous studies have described how calcium signals in the mitochondria convey a message of demand that regulates enzymes to alter ATP production accordingly. This manuscript presents a parallel mechanism that implicates CO2 and $HCO_3$ sensors that regulate cAMP signalling. The authors have identified the putative location of the enzyme pathway following super-resolution imaging of isolated ventricular myocytes and mitochondria. Localizing sAC to the interior portion of the mitochondria is a significant advance and provides a framework for future target discovery.

## Introduction

Cardiac ventricular myocytes work non-stop to meet the blood flow needs of the body, with the heart consuming ATP at high and widely variable rates (*Boyman et al., 2020*; *From et al., 1986*; *From et al., 1990*; *Katz et al., 1989*; *Portman et al., 1989*; *Zhang et al., 1995*; *Elliott et al., 1994*; *Matthews et al., 1981*; *Massie et al., 1994*; *Schwartz et al., 1994*; *Xu et al., 1998*). Because of the everchanging energy needs of the ventricular myocytes and their minimal ATP reserves (*Jacobus, 1985*; *Wang et al., 2010*), ATP production must be dynamically managed by a control system that has high gain and is rapidly responsive (*Maack and O'Rourke, 2007*; *Murphy and Steenbergen, 2021*). Recent work has shown that heart rate-dependent elevation of cytosolic $Ca^{2+}$ signaling drives the ventricular myocytes to contract more frequently while elevating the mitochondrial matrix $Ca^{2+}$ ($[Ca^{2+}]_m$) and thereby increasing ATP production (*Boyman et al., 2020*; *Wescott et al., 2019*; *Garg et al., 2021*). The energetic needs of ventricular myocytes are also linked to local blood supply through the newly described 'electro-metabolic signaling (EMS)'. Through this mechanism, as [ATP] in ventricular myocytes falls with ATP consumption, small blood vessels dilate to increase local blood flow (*Zhao et al., 2020*; *Grainger and Santana, 2020*). EMS thereby will increase the supply of oxygen and nutrient substrates and speed up removal of metabolic waste products by increased flow through local end-arterioles and capillaries. Together, EMS and $[Ca^{2+}]_m$ signaling provide important feedback controls to increase ATP production in working ventricular myocytes. There is, however, still a huge gap in accounting for the full physiological scale of ATP consumption by ventricular myocytes (*Boyman et al., 2020*; *From et al., 1986*; *From et al., 1990*; *Katz et al., 1989*; *Portman et al., 1989*; *Zhang et al., 1995*; *Elliott et al., 1994*; *Matthews et al., 1981*; *Massie et al., 1994*; *Schwartz et al., 1994*; *Xu et al., 1998*). There is no characterized feedback signal yet that provides a mechanism for ventricular myocytes to scale-up ATP production by their mitochondria when increased work is being done by the heart at constant $[Ca^{2+}]_m$. This kind of situation arises routinely when the heart must pump at the same rate but against a greater afterload. This occurs, for example, when blood pressure rises. The investigation presented in this paper seeks to fill this major gap in our understanding.

Here, we report on an understudied and poorly understood signaling pathway in cardiac mitochondria in which $CO_2$ and soluble adenylyl cyclase (sAC) play pivotal roles. $CO_2$ is generated in mitochondria largely as a waste product originating from processing of energy substrates by the Krebs cycle, therefore reflecting the extent of energy metabolism. This $CO_2$ is dissolved in the local aqueous fraction and is in dynamic aqueous equilibrium with bicarbonate ($HCO_3^-$). Importantly, sAC is activated by bicarbonate but unlike 'transmembrane' adenylyl cyclase (tmAC), sAC is found in solution not in the cellular membranes. When sAC is activated, like tmAC, it generates cAMP as the second messenger (*Buck et al., 1999*; *Litvin et al., 2003*).

The 10-member family of signaling proteins known as adenylyl cyclases (ACs) is exquisitely adaptable and transduces a wide array of biological signals by generating the second messenger cAMP (*Zaccolo et al., 2001*; *Di Benedetto et al., 2021*). The family is best known by the nine members that are incorporated in the plasma membranes of almost every cell type via transmembrane (tm) protein domains. While tmACs are broadly regulated by G proteins in response to hormonal stimuli (*Oldham and Hamm, 2008*; *Syrovatkina et al., 2016*), they are able to maintain specificity and high signal-to-noise ratio through proximity to their targets docked at A-kinase anchoring proteins (AKAPs) (*Scott et al., 2013*; *Zaccolo and Pozzan, 2002*). Further signal compartmentalization is gained by nearby 'firewalls' of phosphodiesterases (PDEs) that hydrolyze cAMP to AMP (*Lomas and Zaccolo, 2014*; *Baillie et al., 2019*; *Mongillo et al., 2004*; *Burdyga et al., 2018*). In contrast, there is only one

mammalian member of the sAC subset, and it is much less studied or understood. Mammalian sACs are structural homologues of the bacterial sACs and also lack a transmembrane domain. Instead, they are confined inside organelles like mitochondria, centrioles, and nuclei (*Zippin et al., 2003*; *Feng et al., 2006*; *Acin-Perez et al., 2009*). Unlike tmACs, sACs achieve high local signal-to-noise ratio by producing cAMP inside the small subcellular volumes that also contain their targets (*Tresguerres et al., 2011*; *Rossetti et al., 2021*), namely protein kinase A (PKA) and 'Exchange Proteins Activated by cAMP' or EPACs (*Schmid et al., 2007*; *Sample et al., 2012*; *Parker et al., 2019*).

A number of recent papers investigated sAC and its activation by bicarbonate seeking to identify what was sensed by sAC, where in the cell the reactions took place and what the purpose of this signaling system was (*Acin-Perez et al., 2009*; *Valsecchi et al., 2013*; *Valsecchi et al., 2014*; *Acin-Perez et al., 2011*). This work by Manfredi et al., although widely cited, is controversial (*Di Benedetto et al., 2021*; *Covian et al., 2014*; *Lefkimmiatis et al., 2013*; *Wang et al., 2016*). These publications concluded that the bicarbonate-activated sAC works within the mitochondrial matrix to produce cAMP, which activates PKA, which in turn increases ATP production. It was also reported that the role of this signaling system is to sense and respond to 'nutrient availability' (*Acin-Perez et al., 2009*; *Valsecchi et al., 2013*; *Valsecchi et al., 2014*; *Acin-Perez et al., 2011*). Despite seeming straightforward findings, all key mechanistic findings have been disputed by other investigators (*Di Benedetto et al., 2021*; *Lefkimmiatis et al., 2013*; *Covian et al., 2014*; *Wang et al., 2016*). While almost all published investigations agree that elevated bicarbonate leads to an increase in ATP production, they disagree on how this happens. The role of PKA in the matrix is disputed (*Di Benedetto et al., 2021*; *Lefkimmiatis et al., 2013*; *Covian et al., 2014*; *Wang et al., 2016*) as is the matrix localization of native sAC (*Covian et al., 2014*; *Wang et al., 2016*). Furthermore, key pharmacological tools used in the original study were later found to cause acute mitochondrial toxicity rather than selectively inhibit sAC (*Wang et al., 2016*; *Di Benedetto et al., 2013*). In light of these major points of disagreement, we investigated sAC quantitatively with high spatial resolution to determine where sAC was located, how it worked, where the intermediate signals were generated, and importantly, what its overall role in cellular metabolism may be. Additionally, we decided to use fresh and functional heart tissue as our source of mitochondria because it is one of the most metabolically dynamic tissues. Additionally, we reasoned that by discovering how sAC worked in cardiac mitochondria, we were likely to provide a solid physiological framework to understand how sAC worked in other tissues. In contrast, the original investigations (*Acin-Perez et al., 2009*; *Valsecchi et al., 2013*; *Valsecchi et al., 2014*; *Acin-Perez et al., 2011*) of sAC used mitochondria from cells in which mitochondrial ATP production normally operates over a very narrow range (*van Dyke et al., 1983*; *Bracht et al., 2016*; *Shadrin et al., 2015*; *do Nascimento et al., 2018*; *de Medeiros et al., 2015*; *Colturato et al., 2012*) and sAC signaling itself has a low signal-to-noise ratio. To augment the signal, transgenic sAC was overexpressed in the model cells thereby possibly obscuring the role of native sAC.

When each element was studied, we found that all the key conclusions of the initial investigation by Manfredi et al. linking sAC and PKA in the matrix to ATP production were disputed by our data and those of other investigators (see Table in *Supplementary file 1*). Additionally, we found that sAC signaling reports on nutrient *consumption* not on nutrient 'availability' (*Acin-Perez et al., 2009*; *Valsecchi et al., 2013*; *Valsecchi et al., 2014*; *Acin-Perez et al., 2011*). Furthermore, a major finding of the work presented here is that mitochondrial sAC signaling works independently of, yet in conjunction with, mitochondrial $Ca^{2+}$ signaling system. Together they report on the full scale of ATP needs in health and disease. This resonant or complementary signaling between the two systems enables the mitochondria in cardiac ventricular myocytes to produce ATP continuously at levels appropriate for the physiological demands of the heart.

## Results

An investigation is presented that centers on where sAC is located within the ventricular myocyte, how that position affects its function and how bicarbonate increases ATP production through sAC. That investigation requires high-resolution imaging and quantitative biochemical investigations to determine where sAC resides and what its product, cAMP, controls. This work then examines how the local potential effectors of cAMP – specifically EPAC1 and PKA – may be connected to the dynamic control of ATP production by mitochondria. Under these conditions, sAC signaling is examined in context of

$Ca^{2+}$ signaling to examine their interactions. To place the results in a physiological context they are discussed in the setting of the working heart.

## Localization of sAC in ventricular myocytes

Immunofluorescence localization of sAC at high resolution was undertaken using the Zeiss Airyscan 880 super-resolution microscope and the R21 monoclonal anti-sAC antibody (*Zippin et al., 2003*; *Wang et al., 2016*; *Zippin et al., 2013*; *Fazal et al., 2017*; *Liu et al., 2019*) validated in sAC knockout mice (*Chen et al., 2013*). *Figure 1A* shows simultaneously acquired images of a ventricular myocyte co-labeled for sAC, ATP synthase (complex-V or $C_V$ of the Electron Transport Chain, or ETC), and F-actin. ATP synthase is a component of the inner mitochondrial membrane (IMM) while F-actin filaments are extramitochondrial. *Figure 1A* (left panel) shows the location of sAC (yellow), with a zoomed-in region in the upper right corner. The sAC labeling clearly localizes to the mitochondria, which appear with punctate features (roughly 1 µm across) occurring in rows between the F-actin-containing (blue) myofilaments (see *Figure 1A*; *Barth et al., 1992*). This conclusion is strengthened by the similarity of the distribution of sAC to that of ATP synthase (red in *Figure 1A*). This point is well illustrated in the merged image. The mitochondrial localization of sAC is also supported by colocalization analysis measured by the Pearson's correlation coefficient between $C_V$ and sAC (*Figure 1B*). Likewise, in *Figure 1C, D*, it is clear that neither protein is colocalized with the contractile filament (F-actin). This examination of the transverse axis of the ventricular myocytes shows that sAC and $C_V$ occur at the same location (*Figure 1C*) and the same frequency in the myocyte. While F-actin is also found at the same frequency as sAC and $C_V$, F-actin is only found between them (*Figure 1D, E*). While sAC and ATP synthase are co-localizing to mitochondria, the question remains, where in the mitochondria is sAC located. There are four possible locations to consider: the outer mitochondrial membrane (OMM), the intermembrane space (IMS), the IMM, and the matrix.

To narrow down the possibilities, super-resolution microscopy was applied to isolated mitochondria (*Figure 1F, G*), using immunolabeling for sAC and Tom20 (Translocase of the Outer Membrane 20), an abundant OMM protein (*Omura, 1998*). The resolution achieved clearly distinguished between OMMs of adjacent mitochondria and showed that sAC is located within the perimeter of the OMM but is not co-localized with Tom20. Since sAC does not contain any membrane-spanning sequences, it is unlikely to be embedded in the IMM. This then suggests that sAC is in the IMS or in the matrix or in both compartments.

## Bicarbonate sensing in isolated cardiac mitochondria

There are two important functional distinctions between sACs and tmACs. First, sAC is directly activated by $HCO_3^-$ to produce cAMP while the tmACs are not (*Buck et al., 1999*; *Litvin et al., 2003*). Second, sAC is not activated by forskolin to produce cAMP (*Buck et al., 1999*; *Litvin et al., 2003*), while all nine types of tmACs are (*Zhang et al., 1997*). We used these established features of the different kinds of ACs to determine the identity of the ACs in cardiac mitochondria. As indicated by the experiments of *Figure 2A*, treating isolated mitochondria with bicarbonate robustly activates cAMP production (from 0 to physiological bicarbonate of 15 mM) while treatment with forskolin (25 µM) had no effect on cAMP production. The simple conclusion from these experiments is that the cardiac mitochondria contain functional sAC and no detectable tmACs.

Further definition of the behavior of the mitochondrial sAC system was provided by use of the membrane permeable PDE inhibitor IBMX, 3-isobutyl-1-methylxanthine (*Wells and Miller, 1988*). PDEs are plentiful in nearly all cells that use cAMP and are distributed to help focus the action of the cyclic nucleotide to a local region (*Baillie et al., 2019*; *Maurice et al., 2014*). The PDEs limit the diffusion of the cAMP away from their intended target. This has often been likened to a 'firewall' against the excessive signaling of cAMP within a region of a cell or within an organelle (*Lomas and Zaccolo, 2014*; *Baillie et al., 2019*; *Mongillo et al., 2004*; *Burdyga et al., 2018*). Conversely, inhibition of endogenous PDEs within mitochondrial compartments would be expected to enhance cAMP signaling within those compartments. The effects of the PDE inhibitor IBMX on the action of sAC in isolated mitochondria are presented in *Figure 2B, C*. Specifically, physiological concentrations of $HCO_3^-$ (10 and 15 mM) significantly increase ATP production, and IBMX elevates it further, for a combined effect of doubling ATP output. These data are consistent with a cAMP signaling system inside mitochondria

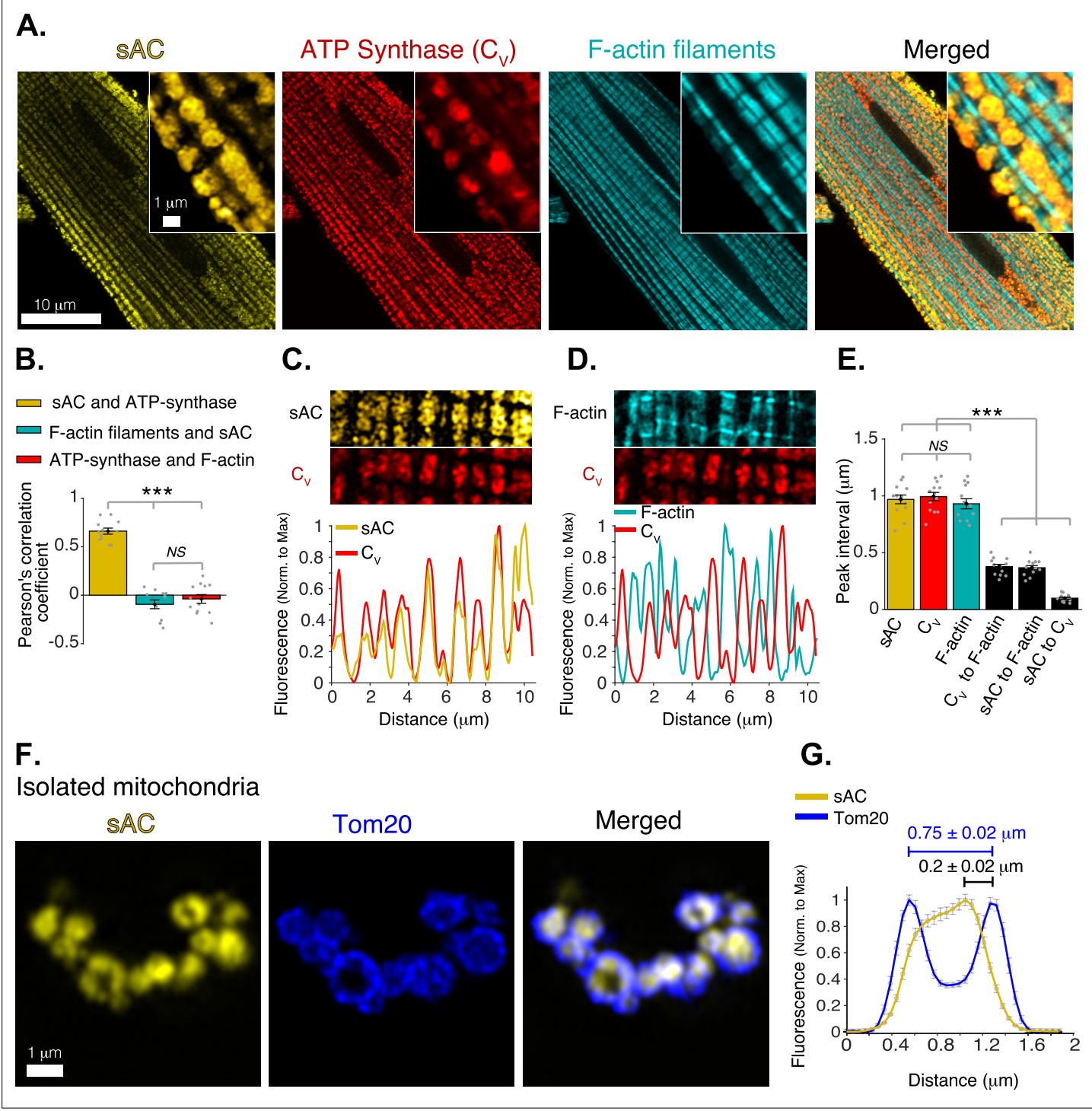

**Figure 1.** Mitochondrial localization of soluble adenylyl cyclase (sAC). (**A**) Airyscan super-resolution fluorescence images of a cardiomyocyte (left to right) immuno-labeled for sAC (yellow) and ATP synthase (red), and loaded with Alexa Fluor 488 phalloidin (1 M) to label the F-actin within the contractile filaments (cyan). Merged image (far right) shows superimposed labeling of sAC, ATP synthase, and F-actin. (**B**) Pearson's correlation analysis of subcellular colocalization of sAC with ATP synthase (yellow), sAC with F-actin filaments (cyan), and ATP synthase with F-actin filaments (red), ($n$ = 12 cells). (**C**) Fluorescence profile of sAC (yellow) and ATP synthase (red). (**D**) Fluorescence profile of F-actin (cyan) and ATP synthase (red). (**E**) Mean of peak interval for data as in (C, D) ($n$ = 13 cells). (**F**) Airyscan super-resolution images of isolated cardiac mitochondria immuno-labeled for sAC (yellow, left), Tom20 (blue, center), and merged images (right). (**G**) Fluorescence profile of sAC (yellow) and Tom20 (blue) for individual mitochondrial images as in F ($n$ = 36). Mean peak intervals (inμm) are indicated ($n$ = 36 mitochondria). Data in (**B, E** and **G**) are mean ± standard error of the mean (SEM). One-way two-tailed analysis of variance (ANOVA) with Bonferroni correction in **B, E**. ***$p < 0.001$. NS = not significant ($p > 0.05$).

*Figure 1 continued on next page*

*Figure 1 continued*

The online version of this article includes the following source data and figure supplement(s) for figure 1:

**Source data 1.** Numeric source data for *Figure 1*.

**Figure supplement 1.** Mitochondrial localization of bicarbonate-activated soluble adenylyl cyclase (sAC).

that responds to bicarbonate by activating sAC, that is modulated by PDEs that reduce cAMP levels, and that regulates the primary function of these mitochondria, the generation of ATP.

The $HCO_3^-$ that activates sAC in mitochondria is produced spontaneously by hydration of $CO_2$, a reaction that is greatly accelerated by the enzyme carbonic anhydrase (CA). As shown in *Figure 2D, E*, we find that cardiac mitochondria contain CA-XIV (also see *Figure 2—figure supplement 1*). As with sAC (*Figure 1F, G*), co-labeling for TOM20 indicates that CA-XIV is localized to an interior compartment of the mitochondria. Thus, the two critical upstream components of the $CO_2$/bicarbonate signaling pathway reside inside the mitochondria of ventricular myocytes. Nevertheless, with the resolution achieved, it cannot be determined whether sAC and CA-XIV are both inside the matrix, the IMS, or both compartments. If sAC and CA-XIV both reside inside the mitochondrial matrix, where $CO_2$ is produced, it would enable sAC to rapidly track changes in $CO_2$ production in terms of $[HCO_3^-]$. However, the alternative possibility in which sAC is in the IMS will also enable effective tracking of $CO_2$ production by sAC despite the low permeability of the IMM to the bicarbonate anion (*Arias-Hidalgo et al., 2016*). Mitochondria in the interfibrillar regions of cardiomyocytes have cristae with extended flat surfaces packed closely together, creating stacks of thin alternating layers of intracristal and matrix spaces (*Figure 2I* and *Picard et al., 2013*). This 'lamellar' membrane morphology combined with the very high permeability of the IMM to $CO_2$ (*Arias-Hidalgo et al., 2016*; *Itel et al., 2012*) is ideal for gas exchange between the two compartments (*Noble, 1983*). In fact, most (85% or more) of the IMS volume in these lamellar mitochondria is in the crista layers adjacent to matrix and not in the peripheral IMS abutting the OMM and cytosol (see legend, *Figure 2I*). Thus, the level of $CO_2$ inside cristae should rapidly adjust in parallel with that in the matrix where it is generated, and spontaneous $CO_2$ hydration should generate steady-state levels of $HCO_3^-$ that track energy consumption. The presence of CA in the same subcompartment as sAC would speed up the response time of the metabolic tracking, but steady-state bicarbonate levels inside cristae should vary with matrix $CO_2$ even in the absence of CA.

## Mitochondrial cAMP signaling

The above results indicate that the cAMP generated inside mitochondria by sAC and regulated by PDEs directly modulates ATP production by mitochondria. To investigate this process in more detail, the characteristics of the sAC signaling system aid us in the design of experiments to quantitatively assess the contributions of sAC to metabolic regulation of the heart. A critical characteristic is that the sAC output signal following activation is cAMP, which cannot readily cross a membrane due to its hydrophilicity (*Lefkimmiatis et al., 2013*; *Di Benedetto et al., 2013*). We can use this information to determine in which compartment sAC signaling is working in ventricular myocytes. In the special case of an isolated mitochondrial preparation, nucleotides like cAMP applied to the extra-mitochondrial space gain entry to the mitochondrial IMS through the numerous VDAC pores in the OMM (*Rostovtseva and Colombini, 1996*; *Rostovtseva and Colombini, 1997*; *Rostovtseva et al., 2002*). However, cAMP has been shown to be excluded from the matrix because the IMM is impermeable to it (*Lefkimmiatis et al., 2013*; *Di Benedetto et al., 2013*). The experiments shown in *Figure 3* use this understanding to investigate sAC signaling in more detail.

*Figure 3A* shows that ATP production in isolated mitochondria with low $[Ca^{2+}]_m$ is small, about 2 nM ATP/mg/s and is not significantly increased by the PDE inhibitor IBMX. These data suggest that there is little or no cAMP present in the mitochondrial compartment in which sAC is located under these conditions. However, when cAMP is added extra-mitochondrially to volume that also contains the mitochondria, there is a clear increase of approximately 50% in ATP production that is further augmented to 100% (doubling) upon addition of IBMX. This finding re-emphasizes that the sAC is in the mitochondria and, moreover, that it is located in the IMS, that is, accessible to externally added cAMP. While nucleotides can enter the IMS through VDAC (*Rostovtseva and Colombini, 1996*; *Rostovtseva and Colombini, 1997*; *Rostovtseva et al., 2002*), it has been shown that cAMP cannot

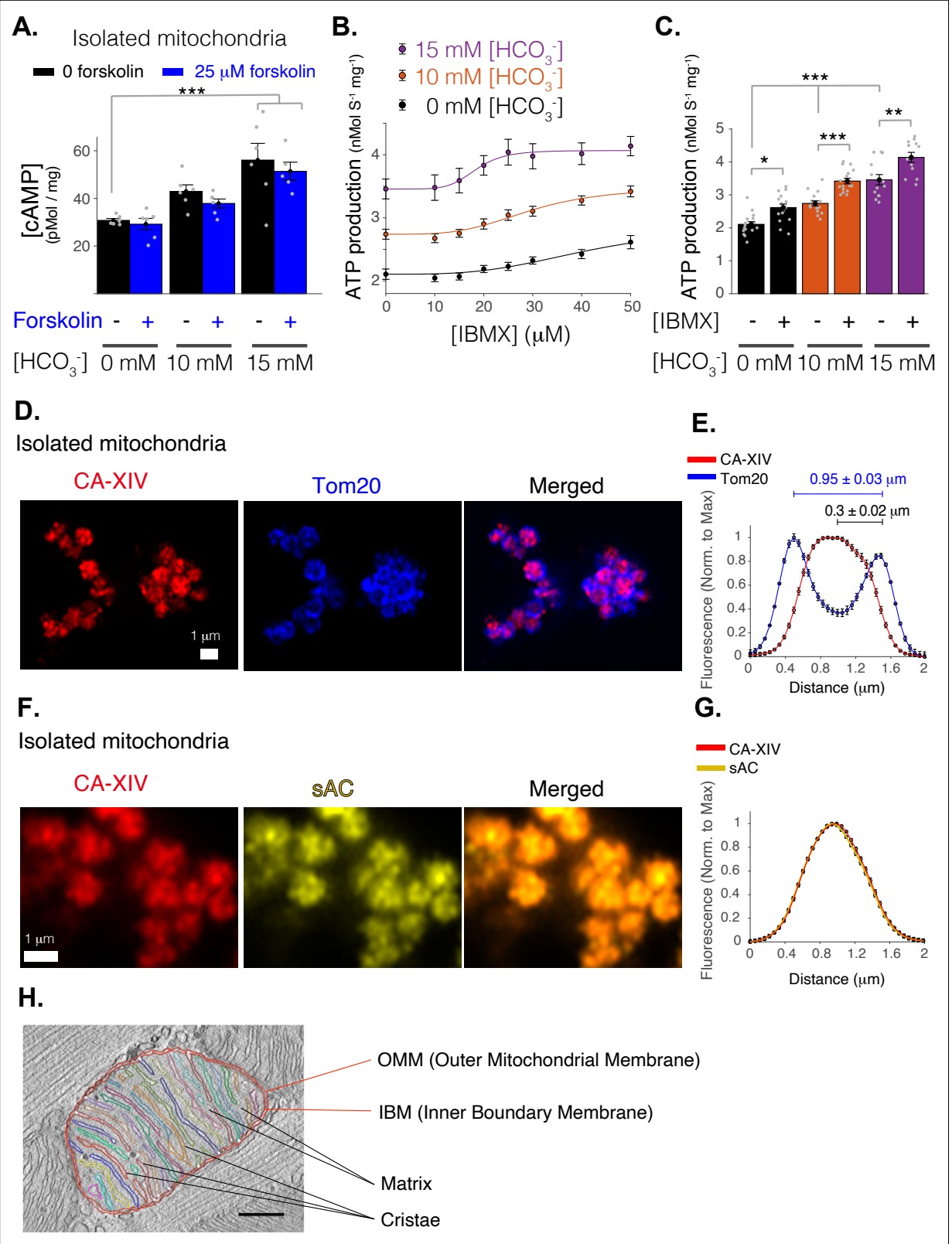

**Figure 2.** Mitochondrial function of soluble adenylyl cyclase (sAC). (**A**) Quantitative ELISA measurements of cAMP inside isolated mitochondria (pmol/mg). cAMP is measured following treatment with indicated concentrations of bicarbonate (HCO₃⁻), which activates sAC, and forskolin, which activates transmembrane adenylyl cyclases (tmACs) (n = 6, 6, 5 for 0, 10, and 15 mM [HCO₃⁻], respectively, mitochondria are isolated independently from four hearts). (**B**) The dependence of ATP production on [HCO₃⁻] and [IBMX]. Sigmoidal fits to the data are shown. (**C**) The sensitivity of ATP production

*Figure 2 continued*

to phosphodiesterase inhibition (with 50 µM IBMX) at different concentrations of $[HCO_3^-]$. (**D**) Airyscan super-resolution images of isolated cardiac mitochondria immuno-labeled for CAXIV (red, left), Tom20 (blue, center), and merged images (right). (**E**) Fluorescence profile of CAXIV (red) and Tom20 (blue) for individual mitochondrial images as in D. Mean peak intervals (in µm) are indicated ($n$ = 32 mitochondria). (**F**) Airyscan super-resolution images of isolated cardiac mitochondria immuno-labeled for CAXIV (red, left), sAC (yellow, center), and merged images (right). (G) Fluorescence profile of CAXIV (red) and sAC (yellow) for individual mitochondrial images as in F ($n$ = 32). For (**B, C**), $n$ = 12–16 independent experiments per group, mitochondria are isolated independently from four hearts. Data in (**A–C**) are mean ± standard error of the mean (SEM). One-way two-tailed analysis of variance (ANOVA) with Bonferroni correction in (**A, C**). *$p < 0.05$, **$p < 0.01$, ***$p < 0.001$. (**H**) Structure of a typical mitochondria from a rat cardiomyocyte showing the intimate arrangement and packing of the cristae and matrix and the IMS. The image is a slice from an electron microscopic tomogram with membranes traced for clarity and quantitation. As shown in the accompanying video of a similar tomogram (*Figure 2—video 1*), crista compartments are formed by closely spaced parallel membranes that extend out of the plane of this image. The functional 'intermembrane space' (IMS) consists of the narrow peripheral space between the inner boundary membrane (IBM) and the outer mitochondrial membrane (OMM) *plus* the spaces inside the cristae that connect to the peripheral space through narrow (20–40 nm) crista junctions (represented by small white circles) (*Frey and Mannella, 2000*). Stereological analysis (*Smith and Page, 1976*) indicates the cristae contain 85% of the total IMS in this mitochondrion, a value that increases to 91% for a mitochondrion with similar crista packing and an area ×2.5 larger. Scale bar is 250 nm.

The online version of this article includes the following video, source data, and figure supplement(s) for figure 2:

**Source data 1.** Numeric source data for *Figure 2*.

**Figure supplement 1.** Intra-mitochondrial localization of carbonic anhydrase 14 (CAXIV).

**Figure 2—video 1.** Electron tomogram of a mitochondrion in the interfibrillar region of a rat cardiomyocyte.
https://elifesciences.org/articles/84204/figures#fig2video1

cross the IMM (*Lefkimmiatis et al., 2013*; *Di Benedetto et al., 2013*). Thus, the production of cAMP by sACs activates a target in the IMS which, in turn, produces a signaling cascade that can activate one or more targets in the IMM and/or matrix space.

There are two possible direct targets for this locally elevated cAMP – namely, PKA and/or EPAC. To examine these targets, experiments were conducted under conditions where matrix $Ca^{2+}$ levels ($[Ca^{2+}]_m$) were measured quantitatively and kept low (under 200 nM). *Figure 3A* (right panel – black bars) shows that, in the *absence* of added extra-mitochondrial cAMP, blocking PKA (with its inhibitor H89) does not affect ATP production nor does the blocking of EPAC1 by the inhibitor CE3F4. However, when cAMP is applied extra-mitochondrially in low $[Ca^{2+}]_m$ (green bars) there is an increase in ATP production that is inhibited by CE3F4 but not H89. From these experiments, we conclude that mitochondrial EPAC1 is the target protein activated by cAMP applied to isolated mitochondria, and that PKA is not involved. We also conclude that EPAC1 contributes to increased generation of ATP even in low $[Ca^{2+}]_m$ and that this signaling happens within the IMS. We conclude that the signaling is in the IMS because cAMP does not enter the matrix under the conditions of our experiment.

*Figure 3B* shows that at elevated $[Ca^{2+}]_m$ (>2 µM), in the absence of cAMP (blue bars), there is a significant increase (2.6-fold) in ATP production compared to the low $[Ca^{2+}]_m$ control (dashed black line). There is an important further increase in ATP production (1.3-fold – brown bars) when cAMP is added to these mitochondria with elevated $[Ca^{2+}]_m$. These findings suggest that IMS sAC signaling and the $[Ca^{2+}]_m$ signaling system have additive (independent) effects on ATP production. This conclusion is also supported by our findings that the sAC signaling system works without the need for elevated $[Ca^{2+}]_m$ (*Figures 2 and 3*). In addition, as shown in *Figure 3A, B*, the cAMP-dependent increase in ATP production was sensitive to EPAC1 inhibition. The pull-down and subsequent immunoblot data in *Figure 3C, D* show that this action can be attributed to activation of the canonical EPAC1 target protein, a member of the Ras-related protein family, Rap1 (Repressor/activator protein 1). Diagrammatically this activation process is shown in *Figure 3E*. While we have shown that much of the signaling (sAC, EPAC1, and Rap1) occurs in the IMS, the exact location and identity of the Rap1-target protein(s) remains unknown and will be examined in future studies.

## cAMP signaling domains: IMS versus matrix

Given the immunolocalization results shown in *Figure 1* using both isolated ventricular myocytes and isolated mitochondria and the functional experiments on isolated mitochondria in *Figures 2 and 3*, the sAC -> EPAC1 -> Rap1 signaling pathway appears to be in the IMS. In contrast to our simple results, other studies have suggested significantly more complex results that conflict with our findings. For example, some experiments suggest that cAMP signaling includes both cytosolic

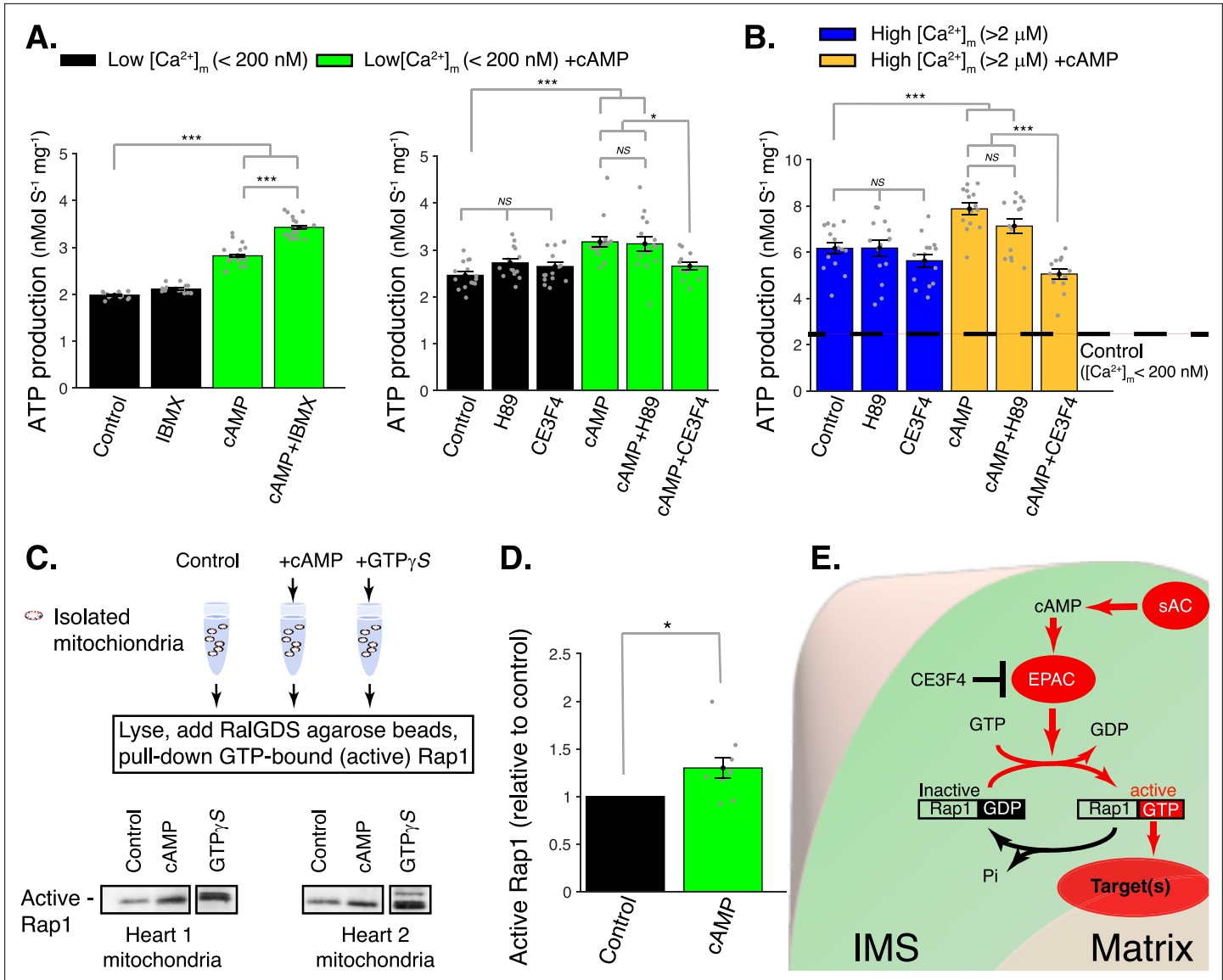

**Figure 3.** cAMP control of mitochondrial ATP production. (**A**) Left, sensitivity of mitochondrial ATP production to cAMP and phosphodiesterase inhibition (with 50 µM IBMX). Measurements are carried out at low $[Ca^{2+}]_m$ (<200 nM). Right, sensitivity of mitochondrial ATP production to cAMP, EPAC1 inhibition (25 µM CE3F4), and protein kinase A (PKA) inhibition (1 µM H89). Measurements were carried out at low $[Ca^{2+}]_m$ (<200 nM). (**B**) Same as A but at high $[Ca^{2+}]_m$ (>2 µM). (**C**) Pull-down and immunoblot analysis for the active form of Rap1 (GTP-bound Rap1) in isolated mitochondria stimulated with cAMP. γ-GTP is used to maximally activate Rap1. (**D**) The relative amounts of active Rap1 (mitochondria isolated independently from $n$ = 9 hearts). (**E**) Schematic diagram showing the likely locations of key proteins in soluble adenylyl cyclase (sAC) mitochondrial signaling. While sAC is clearly located within the intermembrane space (IMS), and both EPAC and Rap1 have signaling domains in the IMS, it is uncertain how the 'target' protein(s) are activated to increase ATP production. For (**A, B**), $n$ = 12–16 independent experiments per group, mitochondria are isolated independently from four hearts. In (**D**), mitochondria are isolated independently from nine hearts. Data in (**A, B**) and (**D**) are mean ± standard error of the mean (SEM). One-way two-tailed analysis of variance (ANOVA) with Bonferroni correction in (**A, B**). One-sample $t$-test for (**D**). *$p$ < 0.05, **$p$ < 0.01, ***$p$ < 0.001.

The online version of this article includes the following source data for figure 3:

**Source data 1.** Western blot analysis in *Figure 3C, D* (anti-Rap1).

**Source data 2.** Original file No. 1 for western blot analysis in *Figure 3C, D* (anti-Rap1).

**Source data 3.** Original file No. 2 for western blot analysis in *Figure 3C, D* (anti-Rap1).

**Source data 4.** Original file No. 3 for western blot analysis in *Figure 3C, D* (anti-Rap1).

**Source data 5.** Original file No. 4 for western blot analysis in *Figure 3C, D* (anti-Rap1).

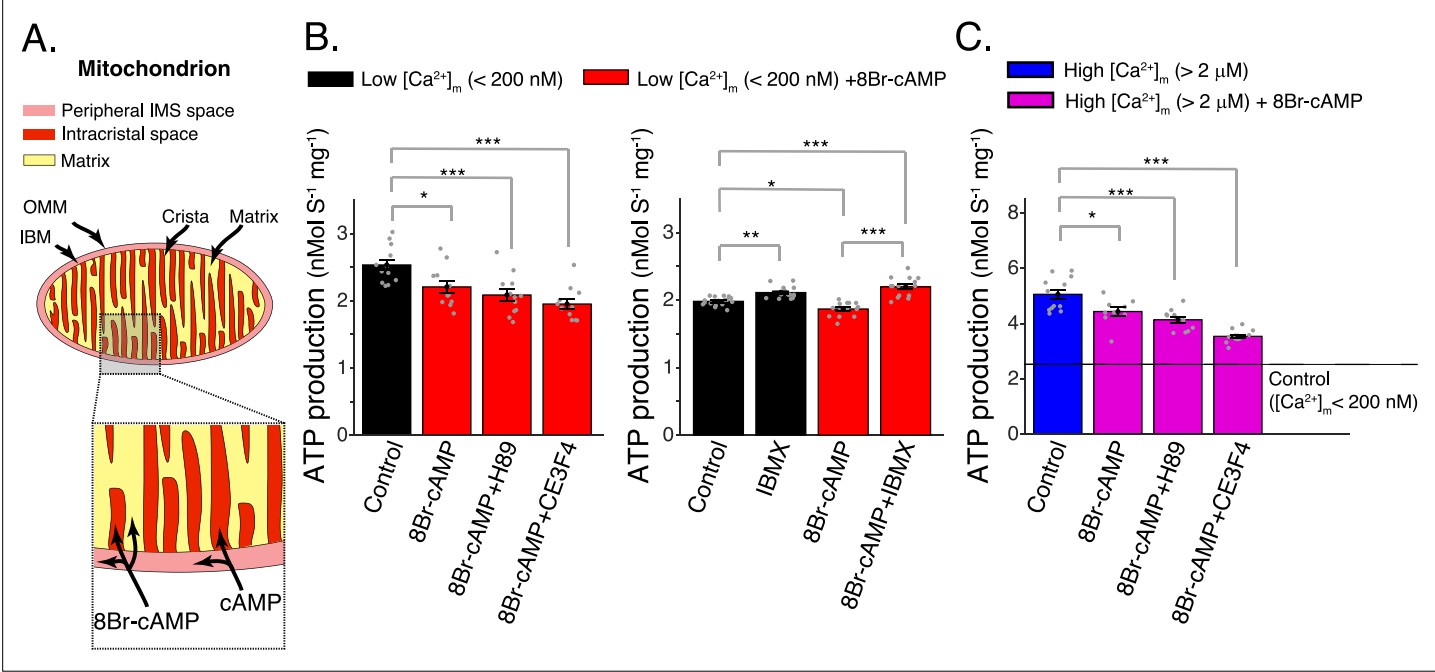

**Figure 4.** Elevated matrix [cAMP] inhibits mitochondrial ATP production. (**A**) Diagram showing mitochondrial compartments accessible to cAMP and the membrane permeant analog, 8Br-cAMP. (**B**) Left, mitochondrial ATP production was decreased by application of 8Br-cAMP. This effect was not changed by either EPAC1 inhibition (25 µM CE3F4) or protein kinase A (PKA) inhibition (1 µM H89). Measurements were carried out at low [Ca$^{2+}$]$_m$ (<200 nM). Right, application of phosphodiesterase inhibitor (50 µM IBMX) produced a small increase in ATP production. Measurements were carried out at low [Ca$^{2+}$]$_m$ (<200 nM). (**C**) Same as (**B**) (left panel) but at high [Ca$^{2+}$]$_m$ (>2 µM). For (**B, C**), $n$ = 12–16 independent experiments per group, mitochondria are isolated independently from four hearts. Data in (**B, C**) are mean ± standard error of the mean (SEM). One-way two-tailed analysis of variance (ANOVA) with Bonferroni correction in (**A–C**). *p < 0.05, **p < 0.01, ***p < 0.001.

The online version of this article includes the following source data for figure 4:

**Source data 1.** Numeric source data for **Figure 4**.

and mitochondrial matrix targets (***Acin-Perez et al., 2009***; ***Acin-Perez et al., 2011***; ***Lefkimmiatis et al., 2013***; ***Di Benedetto et al., 2013***; ***Valsecchi et al., 2017***; ***Kumar et al., 2009***; ***Laudette et al., 2021***). To sort out possible contributions from matrix-localized cAMP signaling, we employed the membrane permeant analog of cAMP, 8Br-cAMP. When applied to an isolated mitochondrial preparation, 8Br-cAMP would be expected to activate both EPAC and PKA in all compartments (i.e., both IMS and matrix) of the mitochondria. 8Br-cAMP also is a persistent activator of EPAC and PKA since it is resistant to breakdown by PDEs (***Christensen et al., 2003***; ***Kawasaki et al., 1998***; ***de Rooij et al., 1998***). ***Figure 4B*** shows the effects of 8Br-cAMP application to isolated mitochondria in low [Ca$^{2+}$]$_m$ (<200 nM). In contrast to the application of cAMP extra-mitochondrially, which produces significant increase in ATP production (***Figure 3***), 8Br-cAMP produces a modest reduction in ATP production. Further, more significant reduction in ATP production is seen with 8Br-cAMP and addition of the PKA inhibitor H89 or the EPAC inhibitor CE3F4 (***Figure 4B***, left panel). When the PDE inhibitor IBMX is applied (***Figure 4B***, right panel), it modestly increases ATP production when [Ca$^{2+}$]$_m$ is low in the absence or presence of 8Br-cAMP, suggesting persistent effects of cAMP produced in the IMS by sAC and undegraded by PDEs under these conditions. The progressive decrease in ATP production following addition of 8Br-cAMP alone and with blockers of PKA (H89) or EPAC (CE3F4) is also observed at the elevated ATP production rates associated with high [Ca$^{2+}$]$_m$ (>2 µM), as shown in ***Figure 4C***. The simple conclusion is that the action of cAMP in the IMS is overwhelmed by the action of 8Br-cAMP in the matrix. Exactly what 8Br-cAMP does in the matrix to produce its effect is not yet known. It is evident, however, that the actions of bicarbonate on sAC, cAMP on EPAC1 and EPAC1 on Rap1 all take place in the IMS. Taken together, the physiological actions of bicarbonate and externally added cAMP that stimulate ATP production by cardiac mitochondria appear to be due to IMS-based signaling, and not to previously described actions of cAMP within the matrix (***Acin-Perez et al., 2009***).

## Discussion

This investigation has addressed a long-standing metabolic conundrum for heart function: how does consumption of energy contribute to the feedback control of ATP production by mitochondria? The $Ca^{2+}$ signaling pathway is clearly important since it links mitochondrial ATP production to heart rate by raising matrix $Ca^{2+}$ levels, which in turn stimulates processes upstream of the respiratory chain (*Boyman et al., 2020*; *Wescott et al., 2019*; *Garg et al., 2021*; *Di Benedetto et al., 2013*; *Luongo et al., 2015*; *Kwong et al., 2015*; *Liu et al., 2021*; *Pan et al., 2013*). Here, we describe a complementary signaling mechanism that further broadens the range of ATP production by mitochondria. We find that cAMP signaling by sAC, likely located in the mitochondrial IMS, appears to set a scale factor for ATP generation in response to increased substrate consumption detected as an increase in the sAC activator, bicarbonate. Soluble AC has two features that make it an ideal metabolic sensor to complement the $[Ca^{2+}]_m$ signal for regulation of mitochondrial ATP production under normal conditions. The first is sAC's distinctive activation by the metabolic waste product $CO_2/HCO_3^-$. In effect, the bicarbonate sensitivity of sAC enables it to 'monitor' the carbon emissions of the ATP production machinery that operate in the nearby mitochondrial matrix. In this way, sAC senses an integrated local energy consumption 'index' that reflects the level of cellular energy demands and $CO_2/HCO_3^-$ production as mitigated by vascular waste product removal. Secondly, sAC generates a powerful, locally acting second messenger, cAMP, as an output signal, based on the $HCO_3^-$ that it senses. Importantly, cAMP as a hydrophilic signaling molecule cannot readily cross the IMM to directly regulate matrix proteins (*Lefkimmiatis et al., 2013*; *Di Benedetto et al., 2013*). Instead, sAC operates within the nanometer scale of the IMS to activate EPAC1 which, in turn regulates guanine-nucleotide exchange factors for the Ras-like GTPase, Rap1. Since neither EPAC1 nor Rap1 has transmembrane domains, the

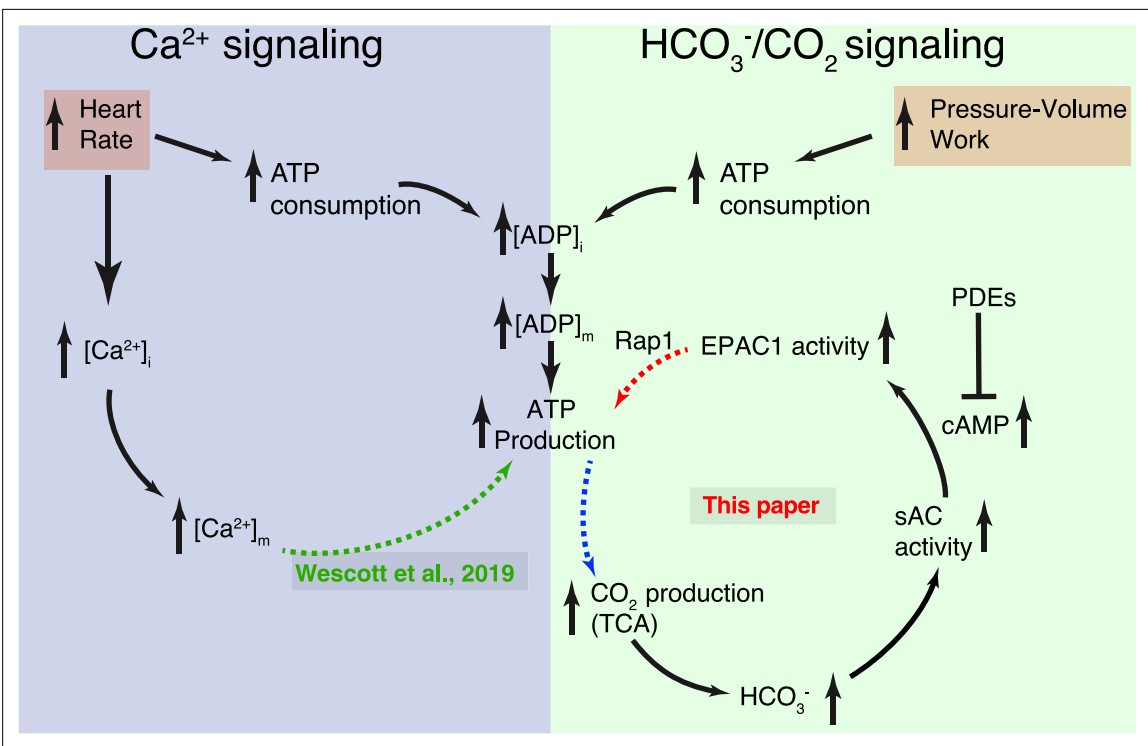

**Figure 5.** Dynamic control of mitochondrial ATP production by $Ca^{2+}$ and bicarbonate. Schematic diagram of the two complementary signaling arms that control mitochondrial ATP production. Left: The $Ca^{2+}$ arm depends on heart rate. With higher heart rates, increased mitochondrial matrix $Ca^{2+}$ ($[Ca^{2+}]_m$) augments ATP production via a mechanism described in detail in *Wescott et al., 2019*. Right: The bicarbonate arm responds to pressure/volume work, highlighting the pivotal role played by $CO_2$ (in dynamic equilibrium with bicarbonate). This signaling arm is centered in the IMS (intermembrane space between the inner mitochondrial membrane [IMM] and the outer mitochondrial membrane [OMM]) with the final link (involving Rap1) reaching across the IMM and probably into the matrix. Neither the $Ca^{2+}$ nor the $HCO_3^-$ signaling arm alone accounts for the full scale of enhancement of ATP production during normal physiological function. Each arm can be stimulated by distinct signals, keeping ATP production in sync with heart rate and pressure–volume work. Importantly, it is the resonance signaling between this two signaling arms that properly matches ATP production to actual energy consumption.

working hypothesis is that Rap1 has an as yet unidentified target in the IMM that affects ATP synthesis either directly or indirectly, by transmitting its signal to another target in the matrix. Importantly, sAC appears to affect mitochondrial ATP production in a manner that does not depend on $[Ca^{2+}]_m$ signaling but instead complements it. In effect, there are two signaling arms for feedback regulation of cardiac ATP production: one that depends on changes in $[Ca^{2+}]_m$ (associated with heart rate) and another that depends on variations in $CO_2/HCO_3^-$ levels (energy consumption), presented schematically in *Figure 5*. These findings provide an important new framework for understanding the dynamic regulation of ATP production in heart and perhaps other tissues. At the same time, the findings raise numerous questions about how the disparate signals are integrated to precisely regulate mitochondrial ATP production over its full dynamic range.

## Bicarbonate signaling in the IMS

Unlike membrane-anchored ACs, soluble ACs are restricted to organelles, whose small volumes confine and amplify cAMP signaling (*Tresguerres et al., 2011*; *Rossetti et al., 2021*). At first glance, the mitochondrial matrix might seem a more likely compartment for a sAC signaling pathway than the IMS. The matrix is the site of $CO_2$ production, and the IMS is more accessible to the highly buffered cytosol. However, it is important to recall that the IMS has two functionally distinct domains. The first is the narrow peripheral region between the 'inner boundary membrane' (IBM) and OMM, which is also termed the 'peripheral IMS'. The second is the crista compartments (intracristal space, ICS) that communicate with the peripheral IMS subregion through small (20–40 nm) crista junctions (*Frey and Mannella, 2000*). In the case of myocardial mitochondria (*Figure 2H*), only ~15% or less of the IMS is peripheral and in close proximity to the porous OMM, while the majority is contained within crista compartments that are functionally distinct from the peripheral IMS region due to restricted diffusion (*Rieger et al., 2014*; *Rieger et al., 2021*; *Garcia et al., 2019*; *Wolf et al., 2019*; *Toth et al., 2020*; *Afzal et al., 2021*). For example, pH gradients of 0.3–0.4 have been detected along the cristae in respiring HeLa cells (*Rieger et al., 2014*; *Rieger et al., 2021*) and between the peripheral IMS and crista interiors in yeast cells (*Garcia et al., 2019*). Such diffusion effects could be even more pronounced for myocardial mitochondria, which have more plentiful, more densely packed lamellar cristae than those of HeLa cells and yeast. As already noted, the interior of myocardial mitochondria is composed of scores of alternating layers of matrix and ICS, causing the cristae to be continuously bathed in the $CO_2$ generated in the matrix. This $CO_2$ would react with water, or 'hydrate', to produce $HCO_3^-$ rapidly (*Garg and Maren, 1972*) such that abrupt rise in $CO_2$ inside the cristae would elevate $[HCO_3^-]$ by several millimolar to its new steady-state levels in just a few seconds (numerical modeling with Stella Professional). Given the rapid kinetics of the hydration reaction when placed in the temporal context of cell function, it seems likely that $[HCO_3^-]$ achieves its equilibrium value inside cristae based on its pH and the partial pressure of $CO_2$ ($pCO_2$) even in the absence of CA. With the newly appreciated geometry of the collection of IMS components (ICS is above 85% of the total IMS), this structure is ideal for rapid gas exchange between the source of $CO_2$, the matrix, and the surrounding layers of ICS.

The proximity of the ICS and IMS layers to the source of $CO_2$ (the adjacent matrix layers) and their restricted communication with the buffered cytosol likely renders the cristae responsive to rapid changes in $pCO_2$ caused by swings in respiration from changes in workload. Furthermore, as we now show in *Figure 2*, the cardiac mitochondria, like other tissues (*Balboni and Lehninger, 1986*), contain CA that accelerate the processes. Thus, as seen by others in HeLa cells, changes in cellular metabolism are associated with changes in pH inside their cristae from 6.7 to 7.4 (*Rieger et al., 2014*) which, assuming $pCO_2$ of 40 mm Hg, spans a range of values for $[HCO_3^-]$ of 5–24 mM that further varies linearly with $pCO_2$ (estimated via the Henderson–Hasselbalch equation). Taken together, bicarbonate-based signaling *could* operate effectively inside the crista micro-compartments of cardiac mitochondria.

## $[Ca^{2+}]_m$ and $CO_2/HCO_3^-$ regulatory mechanisms work together: resonance

Under physiological conditions in heart, increasing workload and heart rate can increase mitochondrial ATP production through $[Ca^{2+}]_m$-dependent signaling (*Boyman et al., 2020*; *Wescott et al., 2019*). In *Figure 5*, this is represented by the dashed green arrow (bottom, left panel). The elevated $[Ca^{2+}]_m$

acts on pyruvate dehydrogenase and glutamate dehydrogenase to increase NADH production and thereby increase proton extrusion through the ETC to hyperpolarize the IMM potential, $\Delta\Psi_M$. As $\Delta\Psi_M$ becomes more hyperpolarized (i.e., more negative), ATP production by the ATP synthase increases (*Boyman et al., 2020*; *Wescott et al., 2019*; *Garg et al., 2021*). In a separate signaling process, increased pressure–volume work increases ATP consumption, elevates ADP levels and promotes substrate oxidation rate. Increased $CO_2/HCO_3^-$ levels activate sAC which generates cAMP and starts a signaling cascade that further increases ATP production by mitochondria (*Figure 5*, circular pathway, right panel). This signaling $CO_2/HCO_3^-$ is produced by the reactions of the TCA (tricarboxylic or Krebs cycle) and pyruvate decarboxylation, both in the mitochondria, as these processes break down and process higher energy substrates. We found that the 'bicarbonate' signaling arm of ATP production has the capacity to double the amount of ATP generated per unit time compared to elevated $[Ca^{2+}]_m$ alone. When considering the two systems working together, there are likely to be many time- and activity-dependent interactions that favor the influence of the $[Ca^{2+}]_m$ arm or the bicarbonate arm of the ATP production machinery. This gives rise to the idea of resonance signaling between the two arms. The immediate and long-term adaptations of the systems are likely to play an important role in health and disease, as well as in physical training and organ-level adaptation of nuclear and mitochondrial gene expression and protein production. Thus, we have broadly presented the notion of the co-existing processes and accompanying cellular and mitochondrial interactions as a kind of 'resonance'. We expect the overall response of ATP production to the two signaling arms (in terms of time and magnitude dependence) to be complex and perhaps not totally independent. Sorting out the control elements and mechanisms that produce the changes will provide a deeper physiological understanding and stimulate novel therapeutic opportunities.

## Myocardial infarction and ATP production mechanisms

Our findings presented in *Figures 1–4* suggest that there is a clear need for the two independent systems to control mitochondrial ATP production – one that depends on $[Ca^{2+}]_m$, and another that depends on $HCO_3^-$. We established these systems by carrying out experiments on ventricular myocytes and mitochondria that were harvested from healthy hearts. However, to broaden our understanding of how these two systems may work together during cardiovascular stress, we also examined mitochondria from diseased hearts. To do this, we measured the functions of both systems in mitochondria taken from 'remodeled' heart tissue in undamaged regions following a myocardial infarction (MI). These results were compared to the same heart region taken from sham operated control hearts. This specific aspect of our discussion is intended to probe the metabolic duality presented in *Figures 1–5* and raise additional points of interest for us and for other investigators. It is by no means part of a comprehensive investigation into remodeling of the heart following MI.

MI was produced by ligation of the left anterior descending (LAD) artery (*Gómez et al., 2001*; *Alvarez et al., 2000*) and examination was done 8 weeks post-surgery. *Figure 6A* shows the large infarct in the anterior wall of the left ventricle. *Figure 6B, C* shows data from sham versus post-MI hearts and reveal a significantly decreased ejection fraction and a significantly decreased fractional shortening of the post-MI left ventricle. Strain analysis is shown in SI (*Figure 2*). Sample tissue used for measurements was taken from the septum and the posterior left ventricle, far from the scar and the MI border zone. The energy demands of these regions were significantly elevated during the 8 weeks following the MI, largely due to compensation for mechanical dysfunction of the scar tissue and the MI border zone. With this approach, our work was directed at testing how mitochondrial ATP production was regulated by $[Ca^{2+}]_m$ and $HCO_3^-$ signaling systems in myocytes with persistently high energetic demands. In these measurements, we found that the mitochondrial ATP production in the absence of cAMP, and at low $[Ca^{2+}]_m$ (i.e., <200 nM), was elevated above the sham control levels. Central protein components of the ATP production machinery were largely unchanged (SI, *Figure 3*). These findings could reflect a change in either regulatory system produced by the post-MI remodeling. It is also possible that both systems had changed because the ATP production that is regulated by $CO_2/HCO_3^-$ and that regulated by $[Ca^{2+}]_m$ are summed at all measurements. In contrast, the ATP production at high $[Ca^{2+}]_m$ (i.e., >2 μM) was elevated to the same levels as control. We also examined the ATP production regulated by $CO_2/HCO_3^-$ (*Figure 6D*) and found that it was increased by application of cAMP. Taken together, these findings are in support of the argument that the $CO_2/HCO_3^-$ system was working as it had been under non-MI conditions. Thus, in all circumstances (i.e., pre-MI and post-MI and sham-MI),

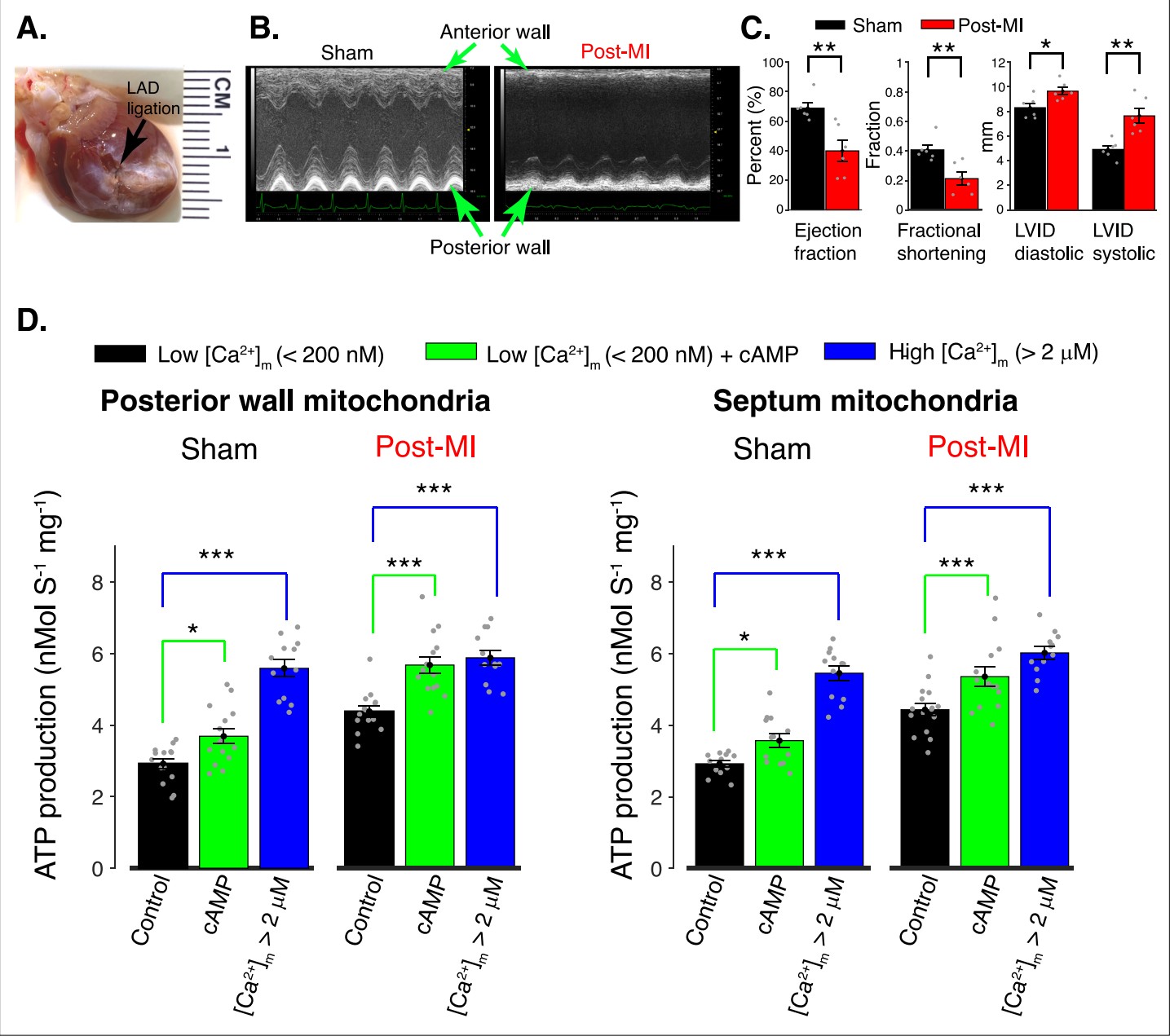

**Figure 6.** Regulation of mitochondrial ATP production in the post myocardial infarction (MI) heart by $[Ca^{2+}]_m$ and [cAMP]. (**A**) Anterior view of rat heart 8 weeks after induction of MI by ligation of left anterior descending (LAD) coronary artery. The black arrow indicates the location of the ligation and the distal, white-colored, region shows scar tissue. (**B**) Parasternal short-axis M-mode echocardiographic images 8 weeks after surgery involving no ligation (Sham) and 8 weeks after genuine LAD ligation (Post-MI). (**C**) Left ventricular echocardiographic measurements from six sham rats (black bars) and from six genuine MI rats (red bars). Data are shown as gray circles. Left chart: Left ventricle ejection fractions and fractional shortening. Right chart: Actual diastolic and systolic left ventricular internal dimensions (LVID). All echocardiographic measurements were carried out under light sedation (1% isoflurane in 100% oxygen). Heart rate (BPM ± SD): sham 318 ± 28, post-MI 317 ± 12. (**D**) ATP production rate in isolated mitochondria at low $[Ca^{2+}]_m$ (<200 nm, black bars), at low $[Ca^{2+}]_m$ plus cAMP (green bars), and at high $[Ca^{2+}]_m$ (>2 μM, blue bars). Mitochondria were isolated from healthy tissue in the septal wall or posterior wall 8 weeks post-MI or post sham operation, as indicated. For all data, $n$ = 12–16 independent experiments per group, four post-MI hearts and four sham hearts. Data in (**C, D**) are mean ± standard error of the mean (SEM). Two-tailed unpaired $t$-test in (**C**). One-way two-tailed analysis of variance (ANOVA) with Bonferroni correction in (**D**). *$p < 0.05$, **$p < 0.01$, ***$p < 0.001$.

The online version of this article includes the following source data and figure supplement(s) for figure 6:

**Source data 1.** Numeric source data for *Figure 6*.

**Figure supplement 1.** Regional contractile failure detected with speckle-tracking based strain analysis.

*Figure 6 continued on next page*

*Figure 6 continued*

**Figure supplement 2.** Central components of mitochondrial ATP production are largely unchanged post-MI.

**Figure supplement 2—source data 1.** Western blot analysis in *Figure 3* (anti-Cv, anti-Civ, anti-Cii, anti-Ci, and anti-TOM20).

**Figure supplement 2—source data 2.** Original file for western blot analysis of posterior wall in *Figure 3* (anti-Cv, anti-Civ, anti-Cii, and anti-Ci).

**Figure supplement 2—source data 3.** Original file for western blot analysis of posterior wall in *Figure 3* (anti-TOM20).

**Figure supplement 2—source data 4.** Original file for western blot analysis of septal wall in *Figure 3* (anti-Cv, anti-Civ, anti-Cii, and anti-Ci).

**Figure supplement 2—source data 5.** Original file for western blot analysis of septal wall in *Figure 3* (anti-TOM20).

both cAMP and $[Ca^{2+}]_m$, are needed to control ATP production (see discussion below) but internal set-points are different.

## Mitochondrial localization and function of sAC

Given the highly localized nature of cAMP signaling (*Scott et al., 2013*; *Zaccolo and Pozzan, 2002*; *Lomas and Zaccolo, 2014*; *Baillie et al., 2019*; *Mongillo et al., 2004*; *Burdyga et al., 2018*) considerable effort was made to pinpoint the sub-mitochondrial localization of sAC. We were able to determine with super-resolution imaging that sAC is predominantly in the interior and not the outer periphery of mitochondria of ventricular myocytes. Subsequent biochemical results, coupled with existing information about the permeability properties of the inner membrane, strongly supports localization of sAC in the mitochondrial IMS where the cAMP that it produces activates EPAC1. In our experiments, cAMP applied in the extramitochondrial solution should passively diffuse into the IMS through the large VDAC pores in the OMM (*Rostovtseva and Colombini, 1996*; *Rostovtseva and Colombini, 1997*; *Rostovtseva et al., 2002*) but cannot permeate the IMM (*Lefkimmiatis et al., 2013*; *Di Benedetto et al., 2013*). Conversely, 8Br-cAMP is a cAMP analog that is broadly used in experiments with intact cells and organelles due to its relatively high membrane permeability (*Christensen et al., 2003*; *Kawasaki et al., 1998*; *de Rooij et al., 1998*). When added to cardiac mitochondria 8Br-cAMP causes ATP production to decline. Assuming the only differences between the signaling effects of cAMP and 8Br-cAMP relate to IMM permeability and insensitivity to PDEs, it appears that elevated matrix cAMP acts to slow down ATP production, overwhelming any IMS-based upregulation mechanism. The lack of stimulation of ATP output by 8Br-cAMP is consistent with findings by Balaban et al. who also used 8Br-cAMP with isolated cardiac mitochondria (*Covian et al., 2014*). However, these results appear to be at odds with those of Manfredi et al. who used 8Br-cAMP with isolated liver mitochondria. They concluded that sAC works *only* in the matrix and not the IMS and that its signaling target is PKA not EPAC (*Acin-Perez et al., 2009*). While it is possible that the discrepancy might be cell-type specific, the finding that elevated matrix cAMP (in the form of 8Br-cAMP) slows down ATP production raises important questions about how mitochondrial cAMP microdomains might operate and interact for metabolic regulation.

## Missing link in the mitochondrial sAC signaling cascade

Mechanistically, we have tracked the signaling sequence that begins with an elevation of $CO_2/HCO_3^-$, followed by AC activation, leading to elevation of cAMP within the IMS, activation of EPAC1 and of Rap1 in the IMS, and eventually to elevation of ATP production by ATP synthase. The localization of these signals shown here for the first time is important as is their link to augmented ATP production. This raises the next (still unanswered) question: What are the mechanistic details explaining how the Rap1 signal leads to increased ATP production? In particular, does a target of Rap1 carry the message to the ATP synthase directly, or is upregulation of ATP generation achieved indirectly, by acting on some component of the ETC or on a critical membrane transport protein such as ANT or VDAC? Any of these mitochondrial components has the potential to increase ATP production if appropriately activated, but thus far, the relevant mitochondrial targets of Rap1 remain unknown.

## Mitochondrial PKA and CaMKII: location and function

Buck and Levine were the first to suggest that sAC resides in mitochondria, based on experiments in HEK293 cells and a novel antibody for sAC (*Zippin et al., 2003*). Manfredi et al. using HeLa, HEK293T, COS-1 cells, and liver mitochondria suggested that sAC was specifically located in the mitochondrial matrix where it activated PKA to affect the electron transport chain (*Acin-Perez et al., 2009*). In

contrast, Balaban et al., using pig heart mitochondria presented evidence that PKA was not working at all within any part of cardiac mitochondria to augment ATP production (*Covian et al., 2014*). In addition, Lefkimmiatis et al. showed that PKA does not affect mitochondria matrix signaling but instead that it associates with the cytosolic facing side of the OMM (*Burdyga et al., 2018*; *Lefkimmiatis et al., 2013*). Here, we have presented findings that broadly do not contradict those of Balaban and Lefkimmiatis, but that extend investigation of cAMP signaling in an important new direction. Unlike previous work of others, we included guanosine diphosphate (GDP) and guanosine triphosphate (GTP) in our experimental buffers to enable the primary catalytic function of guanine-nucleotide exchangers such as EPAC as well as small GTPases such as Rap1 (*Traut, 1994*). Additionally, we used the energy substrate pyruvate instead of glutamate. Pyruvate is not only more substantially metabolized by cardiac mitochondria than glutamate but is also metabolized by a larger number of enzymatic steps, which broadens the search for potentially regulated processes. Moreover, we conducted more direct luminescence measurements of mitochondrial ATP production rate, which also provide higher signal-to-noise ratio than the analogous measurements of oxygen consumption presented in previous studies. With this approach we were able to find that cAMP *does* act to increase ATP production in cardiac mitochondria but only when it operates in the IMS and acts on EPAC not on PKA.

Another signaling pathway in the mitochondrial matrix was reported by Anderson et al. (*Joiner et al., 2012*; *Luczak et al., 2020*). They suggested that mitochondrially localized CaMKII ($Ca^{2+}$/ calmodulin-dependent protein kinase II) has multiple targets within mitochondria. Subsequent work by Lezoualc'h et al. found that mitochondrial CaMKII was activated by EPAC (*Fazal et al., 2017*; *Laudette et al., 2021*). The location of these processes, as presented in these publications, was within the space that includes the IMS, the IMM, and the matrix but was not further specified. Our work clearly supports cAMP-dependent activation of EPAC, and it is possible that CaMKII may also be involved. This will be pursued in future work.

## Summary

Here, we have demonstrated that there are two independent signaling systems that regulate cardiac mitochondrial ATP production under normal conditions and that they complement each other. One is the well-known calcium signaling pathway that we recently redefined (*Boyman et al., 2020*; *Wescott et al., 2019*; *Garg et al., 2021*). The second is described here. Moreover, the details of the operation of this second pathway are presented and tested and, in its new form, constitutes a functionally novel pathway. The major signaling occurs in the IMS and serves to use bicarbonate to monitor carbon-based fuel consumption. That information is translated by sAC to cAMP production and from there to EPAC1 and Rap1 activation. How the signal enters the matrix and details on how it leads to the increased ATP production by the ATP synthase is unknown and under investigation. We show that this cAMP signaling pathway complements the calcium-dependent arm of ATP production in the mitochondrial matrix (*Boyman et al., 2020*; *Wescott et al., 2019*), and that one or both may be modified in cardiac tissue that has been stressed for prolonged periods following MI. It will be important to investigate how each pathway may be modified and contribute to the development of cardiac disease and how they respond to therapeutic interventions.

## Materials and methods
### Mitochondria isolation

Six- to ten-week-old Sprague-Dawley male rats (250–300 g, from ENVIGO, USA, strain code # 002) were anesthetized using Isoflurane (20 min). A thoracotomy and fast excision of the heart was performed, with removal of the atria. The ventricles were minced in ice-cold isolation buffer (IB) containing (in mM): KCl 100, MOPS 50, $MgSO_4$ 5, (ethylene glycol-bis(β-aminoethyl ether)-N,N,N′,N′-tetraacetic acid) EGTA 2, Na-pyruvate 10, $K_2HPO_4$ 10. The minced tissue was washed repeatedly with IB until clear of blood. The remainder of the preparation was conducted in a cold room (4°C). 20 ml of IB containing tissue was transferred to a Potter-Elvehjem grinder and homogenized at high speed for 2 s followed by four repetitive homogenizations with a 1-μm clearance pestle on low speed. The homogenate was centrifuged for 8 min at 600 × *g* after which the supernatant was transferred to a new centrifuge tube. The pellet was resuspended with 10 ml IB and centrifuged for 8 min at 600 × *g*. The second supernatant was pooled with the first and centrifuged again for 8 min at 600 × *g*. The

final supernatant was transferred to a clean centrifuge tube and spun at 3200 × $g$ for 12 min. The supernatant was discarded, and the pellet (the mitochondria sample) was resuspended in resuspension buffer (RB1) base solution containing (in mM): KCl 100, MOPS 50, $K_2HPO_4$ 1, supplemented with Na-pyruvate (10 mM), EGTA (40 µM), and with Fura-2 AM (acetoxymethyl ester form of the calcium indicator Fura-2) (2 µM). After 30 min, the mitochondria were centrifuged at 3200 × $g$ for 12 min and resuspended in RB2, which is RB supplemented with Na-pyruvate (1 mM) and EGTA (40 µM). The mitochondria were centrifuged at 3200 × $g$ for 12 min and a final resuspension and pelleting was done using RB3, consisting of RB and EGTA (40 µM). The concentration of mitochondria was quantified by Lowry assay with a typical rat heart yielding ~15 mg mitochondrial protein. The high purity of mitochondria isolated via this procedure was previously shown (*Wescott et al., 2019*; *Shimada et al., 2022*). Mitochondria were used within 4 hr of isolation. All procedures and protocols involving animal use were approved by the Institutional Animal Care and Use Committee of the University of Maryland School of Medicine (IACUC # 0921015).

## Isolation of ventricular myocytes

Isolated ventricular myocytes were obtained from adult male Sprague-Dawley rats (250–300 g, from ENVIGO, USA, strain code # 002). Animals were euthanized using isoflurane (5%) inhalation anesthesia via a vaporizer. Twenty minutes prior to euthanasia rats were injected with an intraperitoneal heparin bolus (1000 U/kg). A thoracotomy was performed, and hearts were excised during deep anesthesia. Hearts were quickly immersed in ice-cold isolation buffer containing 130 mM NaCl, 5.4 mM KCl, 0.5 mM $MgCl_2$, 0.33 mM $NaH_2PO_4$, 10 mM D-glucose, 10 mM taurine, 25 mM HEPES (4-(2-hydroxyethyl)-1-piperazineethanesulfonic acid) and 0.5 mM EGTA (pH 7.4, adjusted with NaOH). The aorta was cannulated and hearts were then mounted on a Langendorff perfusion system. Hearts were perfused with isolation buffer containing 0.5 mM EGTA buffer for 5 min at 37°C, before perfusion was switched to EGTA-free isolation buffer supplemented with 1 mg/ml collagenase (type II; Worthington Biochemical, USA), 0.06 mg/ml protease XIV, 0.06 mg/ml trypsin, and 0.3 mM $CaCl_2$. After 6–8 min of enzymatic perfusion the heart was dismounted. The ventricles were transferred to isolation buffer supplemented with 2 mg/ml bovine serum albumin (BSA) and 20 mM 2,3-butanedione monoxime and cut into small pieces. Mechanical dissociation with a fire polished Pasteur pipette was performed to achieve further dissolution of the ventricular tissue. The cell suspension was then filtered through a nylon mesh with a pore size of 300 µm. Ventricular myocytes were allowed to pellet by sedimentation, resuspended in NT solution, and were used within 4 hr of isolation. All procedures and protocols involving animal use were approved by the Institutional Animal Care and Use Committee of the University of Maryland School of Medicine (IACUC # 0921015).

## MI model

Male CD rats (175–200 g) underwent surgical ligation of the LAD coronary artery, performed by Charles River surgical services. Following incision between the fourth and fifth intercostal spaces, the LAD was permanently ligated between the pulmonary cone and the left auricle using 5–0 silk sutures. As control cohort, age- and size-matched CD rats underwent a sham procedure that included incision but no coronary artery ligation. Post-surgery, transthoracic echocardiography was carried out every 2 weeks in the MI and sham operated cohorts. All procedures and protocols involving animal use were approved by the Institutional Animal Care and Use Committee of the University of Maryland School of Medicine (IACUC # 0921015).

## Echocardiography

Transthoracic echocardiography was performed using a VisualSonics Vevo 2100. Rats were anesthetized using isoflurane and adjusted to a heart rate of 350 ± 50 BPM. Body temperature was monitored throughout acquisition. Systolic parameters were obtained from short-axis M-mode scans at the midventricular level, as verified by papillary muscles. Apical four-chamber views were obtained, and diastolic function was assessed by pulsed wave doppler imaging across the mitral valve. B-mode imaging in the parasternal long-axis plane was used to calculate global longitudinal strain with VevoStrain software (Visual Sonics). At the end of the acquisition all rats recovered without issues. All procedures and protocols involving animal use were approved by the Institutional Animal Care and Use Committee of the University of Maryland School of Medicine (IACUC # 0921015).

## Immunocytochemistry

200 µl of the ventricular cell suspension (40,000 per ml) were seeded on glass-bottom dishes (Cell Nest, 801001) coated with ECM gel (Sigma, E1270), and allowed to settle for 40 min. Cells were fixed with ice-cold methanol for 5 min and left to air dry. Cells were then washed with phosphate-buffered saline (PBS) 3 × 5 min prior to 2 hr of incubation with blocking buffer (SuperBlock, Thermo #37580). This was followed by overnight incubation (4 °C) with primary antibodies against ATP synthase (Abcam, ab128743, 1:50) and sAC (R21 IHC, CEP BIOTECH, 1:50). Cells were then washed 3 × 15 min with wash buffer (PBS, 0.2% BSA, 0.05% Triton X [vol/vol]) and then incubated with appropriate Alexa Fluor-conjugated secondary antibodies (1:200 in blocking buffer) for 90 min at room temperature. Excess antibodies were removed by washing 3 × 15 min with wash buffer followed by a 3 × 5 min wash with PBS. Cells were then incubated with Alexa Fluor 488 Phalloidin (Thermo, A12379, 1:100 in PBS) for 2 hr. Excess phalloidin was removed by washing the cells 3 × 15 min with PBS.

Isolated cardiac mitochondria were attached to ECM coated glass-bottom 96-well plates by adding 100 µl of 0.01 mg/ml mitochondria to the dish, allowing them to settle for 20 min. Mitochondria were fixed for 20 min in 100 µl 4% paraformaldehyde in PBS, followed by three washes in 200 µl PBS. Membranes were permeabilized utilizing 100 µl 0.05% Triton X-100 in PBS for 15 min. Mitochondria were washed 1× with PBS followed by the addition of 100 µl SuperBlock solution for 2 hr. Following the removal of blocking buffer, primary antibodies were added in 50 µl blocking buffer for 12 hr at 4°C: ATP synthase antibody (Abcam, ab128743 1:200), sAC antibody (R21 IHC, CEP BIOTECH, 1:50), and Tom20 antibody (Proteintech, PTG-11802-AP, 1:200). After primary antibody incubation, mitochondria were washed 3× with 200 µl PBS. Alexa Fluor-conjugated secondary antibodies were added in 100 µl blocking buffer at 1:200 for 2 hr at room temperature. Mitochondria were then washed 3× with 200 µl PBS. For *Figure 2D–G* and for *Figure 2—figure supplement 1a*, isolated cardiac mitochondria were fixed in 4% paraformaldehyde (PFA) for 20 min at room temperature. For Immunohistochemistry (IHC), mitochondria were seeded on glass-bottom dishes (Cell Nest, 801001) coated with ECM gel (Sigma, E1270), and allowed to settle for 40 min. After 1-hr incubation with blocking buffer primary antibodies were added for overnight incubation in 50 µl at 4°C: sAC (R21 IHC, CEP BIOTECH, 1:50), Tom20 (NOVUS BIOLOGICALS, 4F3 H00009804-M01, 1:200), CAII antibody (abcam, ab124687, 1:200), CAVb antibody (ABclonal, A17633, 1:100), and CAXIV antibody (NOVUS BIOLOGICALS, NBP2-98206, 1:100). Dishes were washed 3 × 15 min with wash buffer to remove excess antibodies after which species-appropriate secondary antibodies (Alexa Fluor) were added for 90 min at room temperature. Excess antibodies were removed by washing 3 × 15 min with wash buffer followed by 3 × 15 min wash with PBS. Samples were stored in PBS prior to imaging.

## Airy scan subdiffraction super-resolution imaging

Imaging was carried out with a Zeiss LSM 880 confocal microscope equipped with an Airyscan super-resolution imaging module using a 63/1.40 Plan-Apochromat Oil differential interference contrast M27 objective lens (Zeiss MicroImaging). Laser lines at 488 nm (argon laser), 561 nm (diode-pumped solid-state laser), and 633 nm (HeNe laser) were used to detect Alexa Fluor 488, Alexa Fluor 546, and Alexa Fluor 647, respectively. Imaging with each laser line was carried out sequentially. Z stacks of about 0.54 µm depth with intervals of 180 nm were acquired, followed by 3D Airyscan deconvolution to obtain lateral voxel resolution of ~120 nm (at emission centered at 520 nm). Co-localization analysis was done using ZEN image acquisition and processing software (Zeiss MicroImaging; version 2.3SPL). The Pearson's correlation coefficient ($r$) was used to assess the extent of co-localization between images. Fluorescence intensity profile was analyzed along the transverse axis of ventricular myocytes or the diameter of isolated mitochondria. Fluorescence peak-to-peak distance analysis was carried out using Matlab R2016a.

## Measurements of mitochondrial ATP production and $[Ca^{2+}]_m$

Measurements of mitochondrial ATP production rate and $[Ca^{2+}]_m$ were carried out using a BMG LABTECH CLARIOstar plate reader. Fura-2 AM loaded mitochondria (0.1 mg/ml) were mixed in ATP production assay buffer (AB) consisting of (in mM): K-gluconate 130, KCl 5, $K_2HPO_4$ 1 or 10, $MgCl_2$ 1, HEPES 20, EGTA 0.04, GDP 0.125, GTP 0.25, BSA 0.5 mg/ml, D-luciferin (Sigma) 0.005, luciferase (Sigma; SRE0045) 0.001 mg/ml, pH 7.2. A luminescence standard curve was performed daily over a range of 100 nM to 1 mM ATP with Oligomycin A (15 µM) treated mitochondria. Where

indicated, mitochondria were treated by incubation for 30 min on ice prior to the experiment in AB supplemented with the following: [$HCO_3^-$] (0, 10, or 15 mM, [$Na^+$] was kept constant at 15 mM), 1.25 mM [cAMP], 1.25 mM [8br-cAMP], IBMX (0–50 μM), H89 (1 μM), CE3F4 (25 μM). [$HCO_3^-$] supplemented solutions were made by dissolving [$HCO_3^-$] to 30 mM in AB, the pH was adjusted to 7.2, the solution was rapidly cooled by dilution with 4°C AB to 15 or 10 mM [$HCO_3^-$], kept in close tube on ice, and used for up to 30 min. All incubations in [$HCO_3^-$] supplemented solutions were done on ice and in close containers. The mitochondria were supplemented for 2 min prior to the start of the assay with $Ca^{2+}$ (0–50 μM added) and 1 mM pyruvate and 0.5 mM malate. Assays were initiated by injection of 100 μl ADP (0.05 or 1.0 mM) and luciferin/luciferase in AB to bring the final volume to 200 μl. [$HCO_3^-$] (10 or 15 mM) was dissolved in luciferin/luciferase in AB, kept in closed tube, and used within 5 min. Luminescence signal was recorded for 20 s with 1-s integration. In the absence of ADP only ~10 nM ATP was present in the system. An automated sequence was used to assess each well first for luminescence then subsequently for fluorescence. ATP production rates were scaled to nmol ATP per sec per mg mitochondrial protein (nmol/S/mg). [$Ca^{2+}$]$_m$ was measured via Fura-2 fluorescence ratio, $R_{F2}$ (excitation: 335 ± 6 nM, emission: 490 ± 15 nm/excitation: 380 ± 6 nm, emission: 490 ± 15 nm). The quantitative [$Ca^{2+}$]$_m$ values were obtained according to the following equation:

$$[ca^{2+}]_m = \frac{K_{d,F2m}\beta[R_{F2}-R_{F2,Min}]}{[R_{F2,Max}-R_{F2}]}$$

where $K_{d,F2m}$ = 0.26 μM was obtained as described previously (*Wescott et al., 2019*). The $\beta$ ($F_{380,min}/F_{380,max}$) was measured daily (typically 2.5–2.8), as were Fura-2 $R_{Max}$ (340/380 nm) and $R_{Min}$. Purification of luciferase was critical for accuracy in luminescence measurements for ATP. Briefly, purchased lyophilized powder of luciferase (Sigma, SRE0045) was dissolved to 1 mg/ml in 0.5 M Trizma acetate pH 7.5 (Sigma; T1258). The luciferase in solution was centrifuged at 4°C (14,000 × g) in filtered centrifugal tubes until the volume within the centrifugal tubes had reduced by 80% (Amicon Ultra, Millipore, Ireland, molecular cutoff 3 kD). The filtrate was discarded and 0.5 M Trizma acetate solution was added to the luciferase solution at a volume equal to the volume of the discarded filtrate. This wash cycle was repeated eight times. The concentration of the luciferase stock was re-assessed using a NanoDrop 1000 spectrophotometer (Thermo Fisher Scientific), adjusted to 1 mg/ml, and kept at −80 °C. Similarly, critical was the purification of the ADP stocks as described before (*Wescott et al., 2019*).

## ELISA measurements of cAMP

Quantitative cAMP measurements were carried out on isolated cardiac mitochondria (0.5 mg per sample) using cAMP Direct Immunoassay Kit (Fluorometric) (Abcam, ab138880). Mitochondria samples were treated with 0 or 10 or 15 mM [$HCO_3^-$], and 0 or 25 μM forskolin for 30 min, pelleted at 3200 × g for 12 min, and lysed using the lysis buffer. Florescence measurements of horseradish peroxidase (HRP)-cAMP displacement were carried out on the BMG LABTECH CLARIOstar plate reader (excitation: 540 ± 4 nm, emission: 590 ± 4 nm). A standard curve was performed daily over a range of 0.06–2 nmol cAMP (0.3 nM to 10 μM cAMP).

## Immune pull down of Rap1

RalGDS RBD Agarose beads were used to selectively pull-down the active form of Rap1 (Rap-GTP) from isolated cardiac mitochondria (Rap1 Activation Assay Kit, Abcam, ab212011). Samples of mitochondria, 2.5 mg per sample in 2 ml assay buffer, were treated with 0 or 1.25 mM cAMP for 30 min on ice, added with pyruvate (1 mM) and malate (0.5 mM) and ADP (25 μM), incubated for 2 min at room temperature, pelleted by centrifugation at 3200 × g for 12 min, and lysed using the lysis buffer provided with the kit. As positive control, lysed mitochondria samples were treated with 0.1 mM GTPyS (guanosine 5'-O-[gamma-thio]triphosphate) to maximally and irreversibly activate Rap1. Pull down of Rap1-GTP was carried out according to the manufacturer's instructions. The amounts of active Rap-1 were detected by immunoblot analysis of the pull-down product using anti-Rap1 goat polyclonal antibody (Abcam, ab212011).

## Western blot analysis

Isolated cardiac mitochondria or homogenized freshly harvested ventricular tissue (100–200 µl) were lysed in the same volumes of 2× sodium dodecyl sulfate (SDS) sample buffer (Thermo Fisher Scientific) supplemented with 5% β-mercaptoethanol (Millipore) and incubated at 100°C for 10 min, followed by centrifugation at 21,000 × g at 4°C for 10 min. The supernatants were retained, and protein concentrations were measured directly in these SDS–polyacrylamide gel electrophoresis samples using a NanoDrop 1000 spectrophotometer (Thermo Fisher Scientific). Proteins were separated on 4–20% gradient Novex Tris-glycine polyacrylamide gels (Thermo Fisher Scientific) and transferred onto polyvinylidene fluoride membranes (Bio-Rad Laboratories). Membranes were blocked in 5% blocking-grade nonfat dry milk (Bio-Rad Laboratories) in PBS-Tween 20 and incubated with primary antibodies overnight at 4°C, followed by incubation with secondary antibodies for 60 min at RT. Blots were developed with Super Signal West Pico ECL (Thermo Fisher Scientific) and imaged using Amersham Imager 600 chemiluminescence imager (GE Healthcare Life Sciences). Densitometry quantification was carried out using ImageJ 1.53k (https://imagej.nih.gov/ij/). Primary antibodies used for western blotting were anti-Tom20 rabbit polyclonal antibody (Proteintech, PTG-11802-AP; 1:20,000), anti Rap1 goat polyclonal antibody (Abcam, ab212011; 1:1,000), OxPhos Human WB Antibody Cocktail for ETC Complexes: CI-20(ND6), C-II-30(FeS), C-III-Core2, C-IV-II, and C-V-alpha (Thermo Fisher Scientific, 45-8199; 1:2500). Secondary antibodies used were horseradish peroxidase-conjugated anti-mouse (Cell Signaling Technology, CST-7076) or anti-rabbit (Cell Signaling Technology, CST-7074) or donkey anti-goat (Abcam, ab205723).

## Statistics

All results are presented as mean ± standard error of the mean. All test conditions are described in detail in the paper. All data collected from experiments in accordance with the specified conditions are included in the paper. All data collected from each experiment were included in the paper. All experiments were repeated independently with at least three separate sample preparations. All the experiments were preformed using sample sizes based on standard protocols in the field. All experiments require fresh heart tissue, thus statistical analysis was carried out in parallel with experiments to determine when further repetition was no longer required. Statistical analysis was performed using either OriginPro 2018 or Matlab R2016a statistical packages, all with $\alpha = 0.05$. Where appropriate, column analyses were performed using an unpaired, two-tailed $t$-test (for two groups) or one-way analysis of variance with Bonferroni correction (for groups of three or more). Data fitting convergence was achieved with a minimal termination tolerance of $10^{-6}$. p values less than 0.05 (95% confidence interval) were considered significant. All data displayed a normal distribution and variance were similar between groups for each evaluation.

## Electron tomography and morphometric analysis

Ventricular myocytes from adult male Sprague-Dawley rats, obtained as described above, were glutaraldehyde-fixed, osmium tetroxide post-fixed and epon-embedded according to the protocol described by *Huang et al., 2013* by the Microscope Facility of the Johns Hopkins School of Medicine (Baltimore, MD). Sections were cut by ultramicrotomy to thicknesses of 250 or 370 nm, stained with uranyl /lead, and decorated with colloidal gold particles (~10 nm diameter) for alignment of tilt-series images, by the Advanced Electron Microscopy Group at the Wadsworth Center (New York State Department of Health, Albany, NY) (*Mannella, 2001*). A modified JEOL JEM 4000 transmission electron microscope operating at 400 kV and equipped with a GATAN GIF2002 energy filter was used to record images over a tilt range of ±60° at 1° increments, using a CCD camera with a 1024x1024 detector array (pixel size 1.8 nm). Tomograms were computed by weighted back-projection (*Radermacher, 2006*) using SPIDER (*Frank et al., 1996*) and IMOD (*Kremer et al., 1996*) software. Quantitative analyses of membranes and compartments of mitochondria were calculated from manually traced slices from tomograms (as in *Figure 2H*), using grid-based stereology (*Smith and Page, 1976*).

## Acknowledgements

We thank Joachim Buck, Lonnie Levine, and Melania Balbach from Cornell Medical Center for helpful suggestions and their pioneering work on sAC. They have been generous with their resources and

insights. We thank Ana Maria Gomez (Université Paris-Saclay & Inserm, UMR-S 1180) for valuable discussions at the beginning of this project. We thank Chyong-ere Hsieh, Michael Marko, and Zheng Liu (Wadsworth Center, Albany NY) for their skilled work on electron tomography. We also thank Joseph P Kao for valuable discussions on the dynamic chemistry of bicarbonate. This research was supported by American Heart Association grant 15SDG22100002 (LB), 7U19 AI090959 (LB and WJL); Frontiers in Anesthesia Research Award from International Anesthesia Research Society (LB and WJL); R01 GM129584 (MK); University of Maryland Claude D Pepper Center Grant P30 AG028747 (MG); R01 HL142290 (WJL); 5R35GM140822 (WJL); U01 HL116321 (WJL); T32 AR007592, the NIH Interdisciplinary Training Grant in Muscle Biology (AKC).

## Additional information

### Funding

| Funder | Grant reference number | Author |
| --- | --- | --- |
| American Heart Association | 15SDG22100002 | Liron Boyman |
| National Institutes of Health | 7U19 AI090959 | W Jonathan Lederer<br>Liron Boyman |
| Frontiers Foundation | Award from International Anesthesia Research Society | W Jonathan Lederer<br>Liron Boyman |
| National Institutes of Health | R01 GM129584 | Mariusz Karbowski |
| University of Maryland, Baltimore | P30 AG028747 | Maura Greiser |
| National Institutes of Health | R01 HL142290 | W Jonathan Lederer |
| National Institutes of Health | 5R35GM140822 | W Jonathan Lederer |
| National Institutes of Health | U01 HL116321 | Carmen A Mannella<br>W Jonathan Lederer |
| National Institutes of Health | T32 AR007592 | Andrew Kyle Coleman |

The funders had no role in study design, data collection, and interpretation, or the decision to submit the work for publication.

### Author contributions

Maura Greiser, Data curation, Formal analysis, Investigation, Methodology, Writing – review and editing, involved with all experiments; Mariusz Karbowski, Resources, Formal analysis, Investigation, Methodology, Writing – review and editing, contributed molecular tools and was involved in all western immunoblot analysis; Aaron David Kaplan, Resources, Formal analysis, Investigation, Methodology, Writing – review and editing, performed echocardiography measurements and cardiac strain analysis; Andrew Kyle Coleman, Formal analysis, Investigation, Methodology, Writing – review and editing, performed immunofluorescent analysis; Nicolas Verhoeven, Methodology, contributed to western immunoblot analysis; Carmen A Mannella, Conceptualization, Writing – review and editing, contributed to study design, data interpretation, and writing the paper; W Jonathan Lederer, Conceptualization, Writing - original draft, Writing – review and editing, contributed to study design, data interpretation, and writing the paper; Liron Boyman, Conceptualization, Data curation, Formal analysis, Supervision, Funding acquisition, Validation, Investigation, Visualization, Methodology, Writing - original draft, Project administration, Writing – review and editing

### Author ORCIDs

Maura Greiser http://orcid.org/0000-0002-3197-0910

Liron Boyman http://orcid.org/0000-0002-4485-680X

### Ethics

All procedures and protocols involving animal use were approved by the Institutional Animal Care and Use Committee of the University of Maryland School of Medicine (IACUC # 0921015).

### Decision letter and Author response

Decision letter https://doi.org/10.7554/eLife.84204.sa1
Author response https://doi.org/10.7554/eLife.84204.sa2

## Additional files

### Supplementary files
• Supplementary file 1. Mechanistic findings by investigations of sAC and its role in mitochondria.
• MDAR checklist

### Data availability

The data that support the findings of this study are shown within the figures and their source numeric values are included in this publication as supplementary source data tables. Should additional information be requested it will be available from the corresponding author.

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
