## [Editor Report]

Cardiac function is critically dependent on the homeostatic control of ATP so enough energy is provided for a given task and previous studies have described how calcium signals in the mitochondria convey a message of demand that regulates enzymes to alter ATP production accordingly. This manuscript presents a parallel mechanism that implicates CO2 and HCO_3_ sensors that regulate cAMP signalling. The authors have identified the putative location of the enzyme pathway following super-resolution imaging of isolated ventricular myocytes and mitochondria. Localizing sAC to the interior portion of the mitochondria is a significant advance and provides a framework for future target discovery.

---

## [Decision Letter]

**Decision letter after peer review:**

Thank you for submitting your article "Calcium and Bicarbonate Signaling Pathways have Pivotal, Resonating Roles in Matching ATP Production to Demand" for consideration by *eLife*. Your article has been reviewed by 2 peer reviewers, and the evaluation has been overseen by a Reviewing Editor and Benoît Kornmann as the Senior Editor. The following individual involved in the review of your submission has agreed to reveal their identity: Pawel Swietach (Reviewer #1).

Essential revisions:

1) Both reviewers raise a concern about the location of the enzymes in the putative IMS pathway versus the matrix.

2) An effort needs to be made to biochemically confirm the location. A possible experimental solution has been suggested by one of the reviewers. That is, to determine the relative distribution of carbonic anhydrase in both compartments. For example, is the signal to sAC dependent on CO2 diffusing from the matrix and getting converted to bicarb in the IMS, or is bicarb transported out of the matrix? Perhaps a pulse of CO2 could be given to see if it triggers the response, with or without a carbonic anhydrase inhibitor present.

*Reviewer #1 (Recommendations for the authors):*

A revision should address the primary question if the responses in isolated mitochondria are due to pH (and hence PMF etc) or bicarbonate, or both. Efforts need to be undertaken to alter HCO_3_ at constant pH, e.g. by raising CO2 accordingly. If IMS HCO_3_ is to change over a 10 mM range, then pH would also change. This could be measured in intact cells, but I don't believe there has been evidence for this. Finally, the appropriateness of placing an HCO_3_ sensor beyond a partitioning membrane should be revised – to me, it would seem more sensible to guage HCO_3_ in the matrix, as this is a membrane-bound space, wherein HCO_3_ could rise and fall considerably. Once CO2 leaves the mitochondrion and dissipates, its concentration profile is near constant (as you would expect for a rapidly diffusing gas). At constant and well-controlled pH, this predicts the constancy of HCO_3_ and hence the constancy of sAC activity. If sACs are in the IMS, there must be another reason for this, which is not related to monitoring CO2 production.

*Reviewer #2 (Recommendations for the authors):*

1) While the findings are novel and interesting, the essential conclusions about the localization of the components are based on deduction from a comparison of the effects of soluble cyclic AMP versus permeable cyclic AMP. The paper would be strengthened by actually showing that sAC, EPAC, and rap1 are intermembrane localized, using the standard immunoblot approach of sequential stepwise permeabilization of the outer membrane and the inner membrane to ascertain when the proteins are lost from the preparation. As it stands. the superresolution imaging does not lead to a clear conclusion as to the location of the sAC, apart from confirming that it is in mitochondria.

2) Some elements of the SR may remain attached to the isolated mitochondria. Given that EPAC has been implicated in modulating Ca transfer between the SR and mitochondria, it would be worth testing if thapsigargin or cyclopiazonic acid alters the response to cAMP. It is theoretically possible that some calcium may be cycling in a microdomain even at low (200nM) external Ca. This would, more definitively, rule out that the cAMP effect is related to enhanced Ca uptake by the mitochondria.

3) The paper is structured in a strange way, with the results of figure 6 appearing after the discussion started. It is also unclear what this experiment adds to understanding the mechanisms examined in the paper, other than to show that the non-ischemic post-infarct heart is somehow remodeled – but why or how it is altered is not investigated. While overall cardiac function is impaired in the post-infarct model, the importance of enhanced mitochondrial ATP production by the remaining healthy tissue does not seem to add much relevance to the study.

---

## [Author Response]

Essential revisions:1) Both reviewers raise a concern about the location of the enzymes in the putative IMS pathway versus the matrix.2) An effort needs to be made to biochemically confirm the location. A possible experimental solution has been suggested by one of the reviewers. That is, to determine the relative distribution of carbonic anhydrase in both compartments. For example, is the signal to sAC dependent on CO2 diffusing from the matrix and getting converted to bicarb in the IMS, or is bicarb transported out of the matrix? Perhaps a pulse of CO2 could be given to see if it triggers the response, with or without a carbonic anhydrase inhibitor present.

We thank the reviewers and the editor for the positive reviews and helpful suggestions. We appreciate their overall assessment that this work substantially advances our understanding of a major research question -- how cardiac energy supply is regulated. As acknowledged by the reviewers, the structural and functional examination of cardiac ATP production provides strong evidence that sAC activation by bicarbonate inside the mitochondria leads to increased ATP production via a signaling pathway involving EPAC and Rap1. We also demonstrate that this signaling process operates inside the mitochondrial intermembrane space (IMS) that includes both the peripheral IMS and the intracristal space (ICS). We took the reviewers comments seriously and followed their suggestions to seek additional evidence for this localization. Our super-resolution imaging of isolated ventricular myocytes and mitochondria isolated from these cells localized sAC to the interior of the mitochondria. In the revised manuscript, we also found carbonic anhydrase 14 (CA-XIV) inside the mitochondria (new Figure 2 D-G, and Figure 2 supplement 1). Although resolution between 20 nm (best) to 100 nm is not sufficient to distinguish between matrix and IMS (including intracristal) compartments, given the ultrastructure of heart mitochondria (see new Figure 2H), it is clear that both sAC and CA-XIV are abundant and in the interior of the mitochondria. We explain how this strengthens our earlier *functional* evidence for a bicarbonate–activated sAC signaling pathway in the IMS/intracristal space. We acknowledge that additional verification by higher resolution imaging would provide useful validation for this localization, but such investigation is beyond the scope of this paper and not needed, given the functional results. In light of the reviewers’ comments, we strove to ensure that these important points are stated clearly, and that the language is balanced appropriately throughout the paper.

An important addition stimulated by the reviewers is a discussion of CO_2_ and bicarbonate signaling in the IMS/ Intracristal space. This is now provided in the results and Discussion sections of the revised paper.

Reviewer #1 (Recommendations for the authors):A revision should address the primary question if the responses in isolated mitochondria are due to pH (and hence PMF etc) or bicarbonate, or both. Efforts need to be undertaken to alter HCO_3_ at constant pH, e.g. by raising CO2 accordingly. If IMS HCO_3_ is to change over a 10 mM range, then pH would also change. This could be measured in intact cells, but I don't believe there has been evidence for this. Finally, the appropriateness of placing an HCO_3_ sensor beyond a partitioning membrane should be revised – to me, it would seem more sensible to guage HCO_3_ in the matrix, as this is a membrane-bound space, wherein HCO_3_ could rise and fall considerably. Once CO2 leaves the mitochondrion and dissipates, its concentration profile is near constant (as you would expect for a rapidly diffusing gas). At constant and well-controlled pH, this predicts the constancy of HCO_3_ and hence the constancy of sAC activity. If sACs are in the IMS, there must be another reason for this, which is not related to monitoring CO2 production.

These recommendations have been addressed in the revised sections on pH control, mitochondrial organization, and physiological relevance, as described above. We feel the more detailed discussion about cardiac mitochondrial ultrastructure, spurred by this reviewer’s critique, have strengthened our hypothesis about physiological relevance of the signaling pathway.

Reviewer #2 (Recommendations for the authors):1) While the findings are novel and interesting, the essential conclusions about the localization of the components are based on deduction from a comparison of the effects of soluble cyclic AMP versus permeable cyclic AMP. The paper would be strengthened by actually showing that sAC, EPAC, and rap1 are intermembrane localized, using the standard immunoblot approach of sequential stepwise permeabilization of the outer membrane and the inner membrane to ascertain when the proteins are lost from the preparation. As it stands. the superresolution imaging does not lead to a clear conclusion as to the location of the sAC, apart from confirming that it is in mitochondria.

The biochemical immunoblot approach of sequential stepwise permeabilization of the outer membrane and the inner membrane is an approach which could strengthen the paper and we were attracted to it. We collaborated with experts using these techniques in other cell types but failed to find the optimized conditions that could enable us to succeed. As part of this effort, we learned from our colleagues that such procedures for cardiac mitochondria vary between extremely challenging to nearly impossible. There are, for example, a huge array of possible permeabilizations, extractions, and proteolytic conditions. The sticky membranes and tight packing of the components of the cardiac mitochondria make the methodology exquisitely challenging. Nevertheless, thank you for the suggestion!

2) Some elements of the SR may remain attached to the isolated mitochondria. Given that EPAC has been implicated in modulating Ca transfer between the SR and mitochondria, it would be worth testing if thapsigargin or cyclopiazonic acid alters the response to cAMP. It is theoretically possible that some calcium may be cycling in a microdomain even at low (200nM) external Ca. This would, more definitively, rule out that the cAMP effect is related to enhanced Ca uptake by the mitochondria.

We measured matrix levels of Ca^2+^ ([Ca^2+^]_m_) in our experiments quantitatively using Fura 2-AM. Thus, where indicated, the level of [Ca^2+^]_m_ is below 200 nM. These measured levels are intramitochondrial calcium levels not extra-mitochondrial calcium levels. We can thus rule out the possibility that the actions of cAMP on ATP production is related to enhanced calcium uptake.

3) The paper is structured in a strange way, with the results of figure 6 appearing after the discussion started. It is also unclear what this experiment adds to understanding the mechanisms examined in the paper, other than to show that the non-ischemic post-infarct heart is somehow remodeled – but why or how it is altered is not investigated. While overall cardiac function is impaired in the post-infarct model, the importance of enhanced mitochondrial ATP production by the remaining healthy tissue does not seem to add much relevance to the study.

We respectfully disagree. While Figures 1-5 report the physiological aspects of this study, Figure 6 raises an additional point for discussion, "Is this bicarbonate driven, sAC-based regulation of ATP production important for energy supply in the failing heart?" Figure 6 shows that the signaling pathway is relevant in post-MI hearts with HFrEF, which underscores the potential clinical importance of the current study in common disease. At the present time we have little understanding of how mitochondria function in the failing heart and how it contribute to this disease. Thus, it is valuable to share new relevant findings that can contribute to better understanding. Planned future studies will seek to better explain the mechanistic underpinnings in normal and failing hearts.